# Intelligent wireless power transfer via a 2-bit compact reconfigurable transmissive-metasurface-based router

Wenzhi Li [1], Qiyue Yu[2], Jing Hui Qiu[1] & Jiaran Qi [1] ✉

With the rapid development of the Internet of Things, numerous devices have been deployed in complex environments for environmental monitoring and information transmission, which brings new power supply challenges. Wireless power transfer is a promising solution since it enables power delivery without cables, providing well-behaved flexibility for power supplies. Here we propose a compact wireless power transfer framework. The core components of the proposed framework include a plane-wave feeder and a transmissive 2-bit reconfigurable metasurface-based beam generator, which constitute a reconfigurable power router. The combined profile of the feeder and the beam generator is 0.8 wavelengths. In collaboration with a deep-learning-driven environment sensor, the router enables object detection and localization, and intelligent wireless power transfer to power-consuming targets, especially in dynamic multitarget environments. Experiments also show that the router is capable of simultaneous wireless power and information transfer. Due to the merits of low cost and compact size, the proposed framework may boost the commercialization of metasurface-based wireless power transfer routers.

The Internet of Things (IoT) is an emerging realm that aims at interconnection with every daily object without human interaction[1–3], which has immense potential to improve many aspects of human life. The IoT devices are widely used for medicine, industry, and smart homes[2,4]. Envisioned grandiose deployments require IoT with self-configuring capability especially with the development of wireless communication. However, there are obstacles and difficulties for conventional battery supply[5], such as environmental protection and use interruption. Therefore, long-term and sustainable power delivery to devices without use interruption is of great benefit for realization of IoT. Wireless power transfer (WPT) is a promising technique, since it can enable power transfer without cables, bringing new opportunities such as mobility, no wires, and eternal energy supply[6,7]. Ambient and dedicated WPT technologies are developed to accelerate the growth of IoT[8,9]. The ambient WPT enables capturing wireless energy from environments, such as sunlight, wind, thermal energy, and wireless

microwave energy. The availability of the ambient energy usually relies on the time, location, weather, and so on. On the other hand, the dedicated WPT is characterized by delivering wireless energy through the energy network infrastructure, providing a steady energy supply strategy. Recently, near-field WPT (NFWPT) through magnetic-field coupling has been developed[10–13]. Conceptually, two coil resonators, one as transmitter and the other as receiver, both resonating at the same frequency, transfer power through non-radiative field. Despite those successful NFWPT experiments, the coupling strength is sensitively dependent on the relative positions of the two coil resonators[11–13], which limits their applications to multiple targets over long distances[14].

With the development and popularization of wireless communication, portable devices like smartphones and small sensors are distributed in sophisticated electromagnetic environments. The indeterminacy of the quantity, location, and power consumption of

[1]Department of Microwave Engineering, School of Electronics and Information Engineering, Harbin Institute of Technology, Harbin, China. [2]Department of Communication Engineering, School of Electronics and Information Engineering, Harbin Institute of Technology, Harbin, China. ✉e-mail: qi.jiaran@hit.edu.cn

IoT devices brings new challenges for WPT[5,7]. As the energy needs of electronic devices are reduced according to the Koomey's law, farfield WPT (FFWPT) has great potential for low-power wireless applications[15,16]. Phased array is one of extensive research topics of FFWPT[17]. It usually requires hundreds of high-resolution phase shifter, low-noise amplifiers, power amplifiers, and analog-digital convertors. The utilization of phased array usually results in bulkiness, high complexity, and high cost, and thus, it may not be suitable for scenarios with low cost and compactness[18,19].

On the other hand, metasurfaces have seen exponential growth[20,21]. Owing to their remarkable electromagnetic properties, a series of successful demonstrations using metasurfaces have been carried out, such as computational imaging[22], holography[23], nonreciprocity[24], achromatism[25], and nonlinearity[26]. Metasurfaces also open a new paradigm for FFWPT[27]. For example, near-field focusing with metasurfaces has been demonstrated[28–31]. Dynamic metasurface antennas (DMA), one of the metasurface-based WPT technologies, consist of a series of reconfigurable-unit-cells-loaded waveguides connected to a set of radio-frequency (RF) chains[32–34]. Another area of significant interest is programmable metasurfaces[35], which utilize digitally programmable unit-cells to provide a link between the physical and digital worlds[36]. Thanks to their prominent characteristics, programmable metasurfaces have found applications in various domains, including space-time modulations[37–39], polarization controls[40], and meta-imagers[41,42]. In addition, optically driven metasurfaces are proposed, allowing for microwave manipulation using light[43,44]. Self-adaptive metasurfaces have also been extensively studied[45,46], showcasing their potential in many scenarios, such as beam steering[47], cloaks[48], and dynamic reactions[49]. In the quest for energy- and spectral-efficient solutions for wireless communication systems, reconfigurable intelligent surfaces (RIS) have emerged as a vital area of exploration[50–55]. They are envisioned to create reconfigurable wireless environments for the next generation of wireless communication.

Despite these successfully demonstrated metasurfaces, complex environments like indoors and industries cultivate new needs for metasurface-based WPT systems. Firstly, the coexistence of large amounts of IoT devices, such as unmanned delivery cars and small sensors, in sophisticated electromagnetic environments becomes an intractable problem for the WPT systems. Flexible charging strategies for selected power-consuming devices are crucial in environments where multiple devices appear and disappear at random. In addition, in practice, finite spaces necessitate device miniaturization and integration. Conventional feeders used to excite metasurfaces, such as horn antennas, suffer from non-uniform wave fronts, resulting in large system profiles on the order of several to ten wavelengths. The large profiles make them challenging for applications with requirements for miniaturization and integration. Furthermore, with the large deployments of IoT devices, wireless spectrum bands, like unlicensed bands, have seen increasingly crowded usage. Since electromagnetic waves can simultaneously carry energy and information, there is a need for the fusion of WPT systems and wireless information transfer (WIT) systems to take full advantage of the wireless spectrums. This fusion may also help to address the spectrum shortage problem.

In this article, we propose a subwavelength WPT framework, which consists of a planar-wave feeder, a transmissive 2-bit reconfigurable metasurface-based beam generator, an environment sensor, and a computation unit. The plane-wave feeder and the beam generator create a compact reconfigurable power transmitter. The total profile of the two components is 0.8 wavelength. The subwavelength profile is an order of magnitude smaller than that with conventional feeders, making it suitable for system miniaturization and integration. Cooperating with the sensor capturing environmental information, the computation unit is able to control the reconfigurable power router in real-time to deliver wireless power to single or multiple targets in

stochastic environments. Experiments demonstrate that the framework has the capability of detecting and localizing multiple targets, and then selectively supplying power to those targets according to their energy requirements. In addition, we also demonstrate that the proposed framework enables simultaneous wireless information and power transfer (SWIPT), providing a possible remedy for the fusion of WPT systems and WIT systems. The proposed framework holds immense application potential in dynamic environments with wireless devices like sensors, smart home devices, and industrial delivery robots.

## Results

### Framework of the proposed WPT system

The proposed WPT framework with a subwavelength profile is shown in Fig. 1. It consists of four parts: a compact integrated near-field feeder (NFF), a dynamic beam generator (DBG), an environment sensor, and an intelligent computation unit. Different from conventional feeders that generate a non-uniform wavefront in the nearfield, the NFF is designed to generate a uniform wavefront (a plane-wave-like wavefront in the nearfield, as shown in the following section) so as to properly excite the entire DBG. This enables efficient reduction of the system size to a subwavelength scale, which is one order of magnitude smaller than other works (see Supplementary Note 1 for a detailed comparison). The NFF comprises two components: an artificial magnetic conductor (AMC) loaded with a 2 × 2 antenna array and a partial reflective plane (PRP). The AMC and PRP create a Fabry-Perot cavity (FPC). The antenna array feeds the power to the FPC, where the PRP leaks a uniform wavefront. The DBG is a space-time-modulated programmable metasurface that consists of many active transmissive unit-cells. Each unit-cell provides tunable 2-bit local phase by incorporating four PIN diodes. By applying two different voltages to the PIN diodes, we can dynamically change the transmissive phase of each unit cell between four states. The NFF and the DBG combine to form a reconfigurable power router capable of delivering wireless power to multiple power-consuming targets. The environment sensor is applied to gather dynamic information in a situation where the quantity and locations of the power-consuming targets are unknowable (such as smartphone and unmanned devices, as depicted in Fig. 1). The intelligent computation unit processes the information from the sensor and then generates a phase pattern to control the router. In this way, we can deliver wireless power intelligently without the need of human intervention. In the following sections, we will discuss how to implement the NFF and DBG, and how to incorporate the environment sensor with them to realize the proposed WPT framework.

### Design of the proposed WPT system

We first introduce how to design the NFF to generate a uniform excitation. To excite a metasurface, we usually consider two points: (1) illumination on each unit-cell should be as uniform as possible to fully use the entire metasurface; (2) leakage of energy out of the metasurface should be as minimal as possible in consideration of power transfer efficiency. A traditional method of exciting a metasurface is placing a high-gain horn antenna to meet these points. However, such a configuration usually leads to a large system profile, making integration and miniaturization challenging. On the one hand, a high-gain horn antenna usually has a narrow beamwidth, which reduces energy leakage, but requires a large distance away from the metasurface for uniform illumination. In other words, there needs a relatively significant distance between the horn antenna and the metasurface to ensure that the beam spreads widely enough to excite the metasurface uniformly. On the other hand, a low gain antenna may help with profile reduction, but because of its larger beamwidth, it typically suffers from significant energy leakage, which degrades system performance.

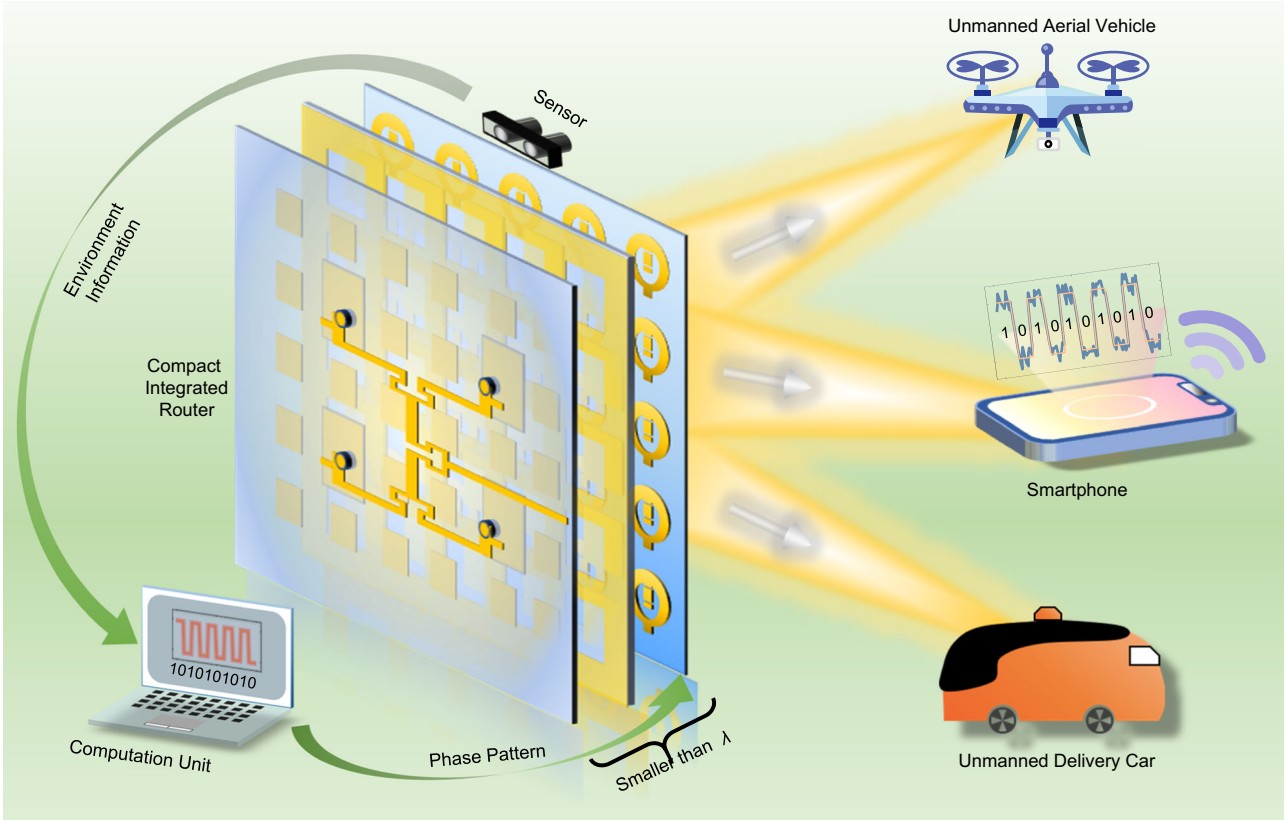

**Fig. 1 | The framework of the proposed WPT system and its photographic scenes.** It consists of a NFF, a DBG, an environment sensor, and an intelligent computation unit. The core components (the NFF and DBG) have a subwavelength profile. This framework enables wireless power transfer to multiple targets in dynamic, complex environments.

To address this contradiction, we design a high gain and low profile NFF as shown in Fig. 2a. The working principle of NFF is inspired by the FPC, which consists of two highly reflective mirrors and is used to generate laser in optical applications. Despite the fact that FPC antennas have been extensively studied in microwave engineering[56,57], the use of FPC antennas as nearfield feeders is rare. In our design, we replace two highly reflective mirrors with an AMC and a PRP. The input power is injected through the 2 × 2 antenna array loaded on the AMC. The injected power is completely reflected by the AMC and then leaks through the PRP. By properly tuning the reflective phases and separation distance of the AMC and PRP, the NFF resonates with the constructive interference of the leaked wave by the PRP and thus generates a uniform nearfield excitation.

We performed the electromagnetic simulation using the commercial software CST Microwave Studio. With the reflective phases of AMC and PRP set as 0° and 175°, respectively, as presented in Fig. 2b, the separation distance is approximately $0.25\lambda$ (see Supplementary Note 2 for details). In the proposed NFF, the utilization of a 2 × 2 antenna array is for uniform excitation of the entire cavity. As shown in Supplementary Fig. S6a, when we use only one antenna for excitation, the output wavefront behaves like a spherical wavefront. Supplementary Fig. S6b shows the output wavefront when two antennas are applied and arranged along $y$ axis. Although the trend of the wavefront in the $xoz$ plane is similar to that in Supplementary Fig. S6a, the wavefront in the $yoz$ plane is flattened, behaving like a plane wave. Supplementary Fig. S6c shows that as the 2 × 2 antenna array is applied, both wave fronts in the $xoz$ and $yoz$ planes become uniform, enabling a planar-wave-like excitation for the DBG.

Figure 2d–h shows the fabricated NFF. The PRP consists of 17 × 17 unit-cells while the AMC consists of 19 × 19 unit-cells. Both of them have a size of 300 mm × 300 mm. The simulated results of nearfield

distribution are shown in Fig. 2i, j. As can be seen, the nearfield phase distribution is nearly flat over the measurement plane, although the phase at the four corners of the NFF degrades. However, it only slightly affects the performance of the entire router. This is because the output power at the corners of the NFF is dramatically lower than the center region, as depicted in Fig. 2i. The corresponding measured nearfield results are shown in Fig. 2k, l. As can be seen, the phase distribution is as flat as the simulated one, and the amplitude is also consistent with the simulation.

Next, we use a 2-bit reconfigurable transmissive metasurface (RTM) to implement DBG for dynamic multiple-beam assembly. Although continuous manipulation of the transmissive phase is more favorable, it is relatively difficult and high-cost to control each unit-cell with an independent and continuously-tunable DC power supply because a metasurface usually consists of hundreds of unit-cells. Therefore, phase quantization is necessary and more suitable for modern digital circuits. We do not use reflective reconfigurable metasurfaces because they have two drawbacks for our low-profile WPT system. First, they can only manipulate power distribution on the same side as feeders, which makes the proposed NFF difficult to deploy. Second, they may cause a blockage effect on the transferred power.

Figure 3a presents the proposed RTM which is composed of 13 × 13 unit-cells with a total size of 300 mm × 300 mm. Each unit-cell consists of two parts: a receiver and a transmitter, which constitute a receiver-transmitter architecture[58–60]. Each part encompasses an elliptical ring antenna, a DC layer and a ground plane. A metal via vertically penetrating through the unit-cell is used to connect the transmitter with the receiver. For the receiver part, the elliptical ring antenna is loaded with two PIN diodes connected by a metal strip passing through the center. The DC layer, composed by two crescent

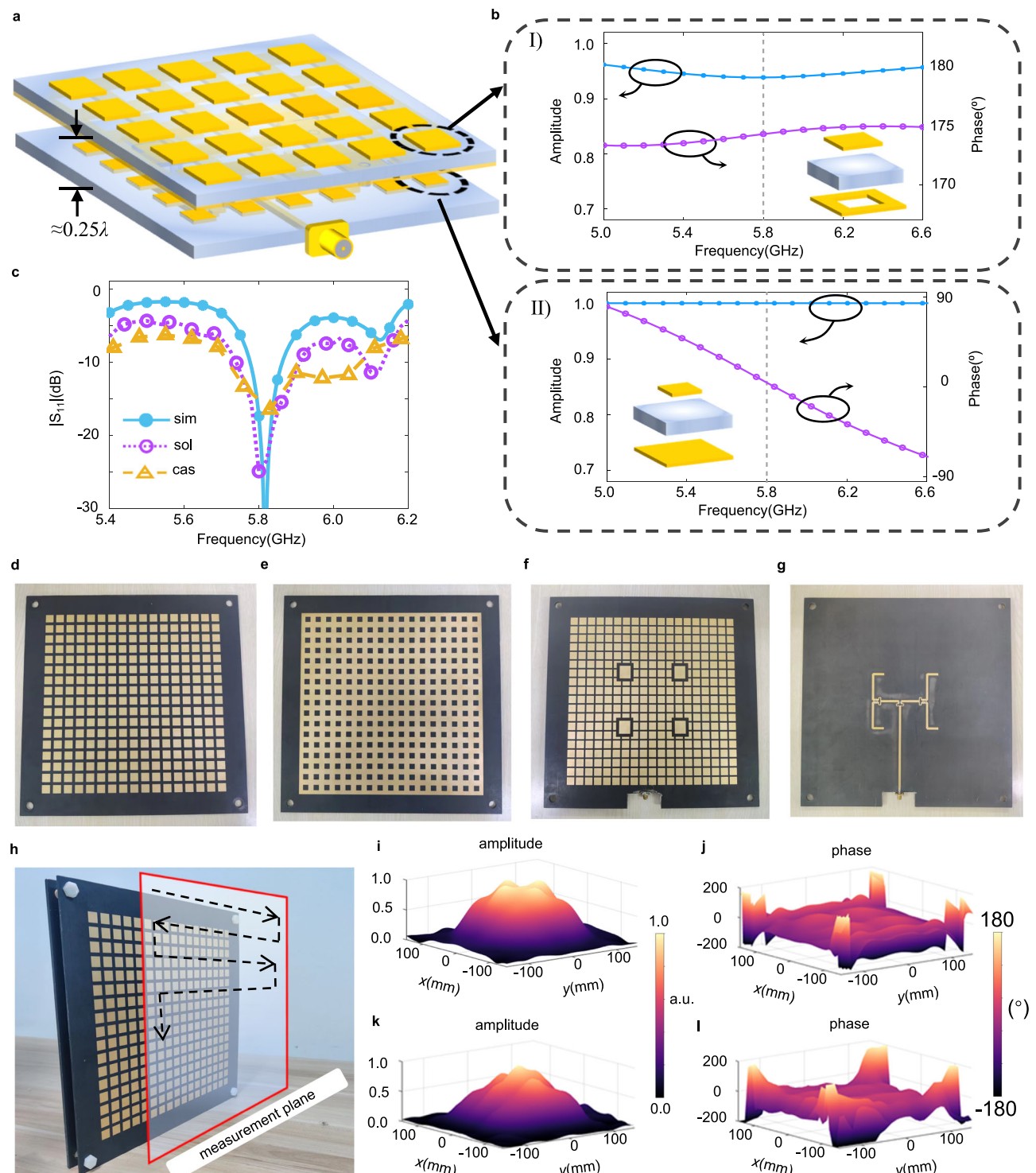

**Fig. 2 | The NFF and its properties. a** The schematic of the proposed NFF (for clear view, the ground plane is transparency). **b** The geometry of the unit-cells of the PRP (I) and AMC (II) with their amplitude and phase responses. **c** The simulated and measured reflection coefficients ($S_{11}$). SIM indicates the simulated $S_{11}$ of the NFF; SOL indicates the measured $S_{11}$ of the NFF; CAS indicates the measured $S_{11}$ of the NFF cascaded with the DBG. **d**, **e** The top (**d**) and (**e**) bottom views of the PRP. **f**, **g** The top (**f**) and (**g**) bottom views of the AMC. **h** The fabricated NFF. The distance of the measurement plane away from the front face of NFF is 50 mm. **i**, **j** The simulated electrical field of amplitude (**i**) and phase (**j**) distributions of the NFF. **k**, **l** The measured electrical field of amplitude (**k**) and phase (**l**) distributions of the NFF. a.u.: arbitrary unit.

metal patch constituting large capacitors with the ground plane, prevents incident high-frequency signals from entering the controlling circuit board. Two symmetric metallized vias connect the elliptical ring antenna with the DC layer for electrical connection and keeping current balance on the elliptical ring antenna. Switching the states of PIN

diodes changes the current flowing direction: for ON state, the PIN diode can be regarded as low impedance component; otherwise, it is a high impedance component (see Supplementary Note 3 for the detailed equivalent circuit model). For example, when the Diode#1 is OFF and Diode#2 is ON, incident wave induces current that mainly

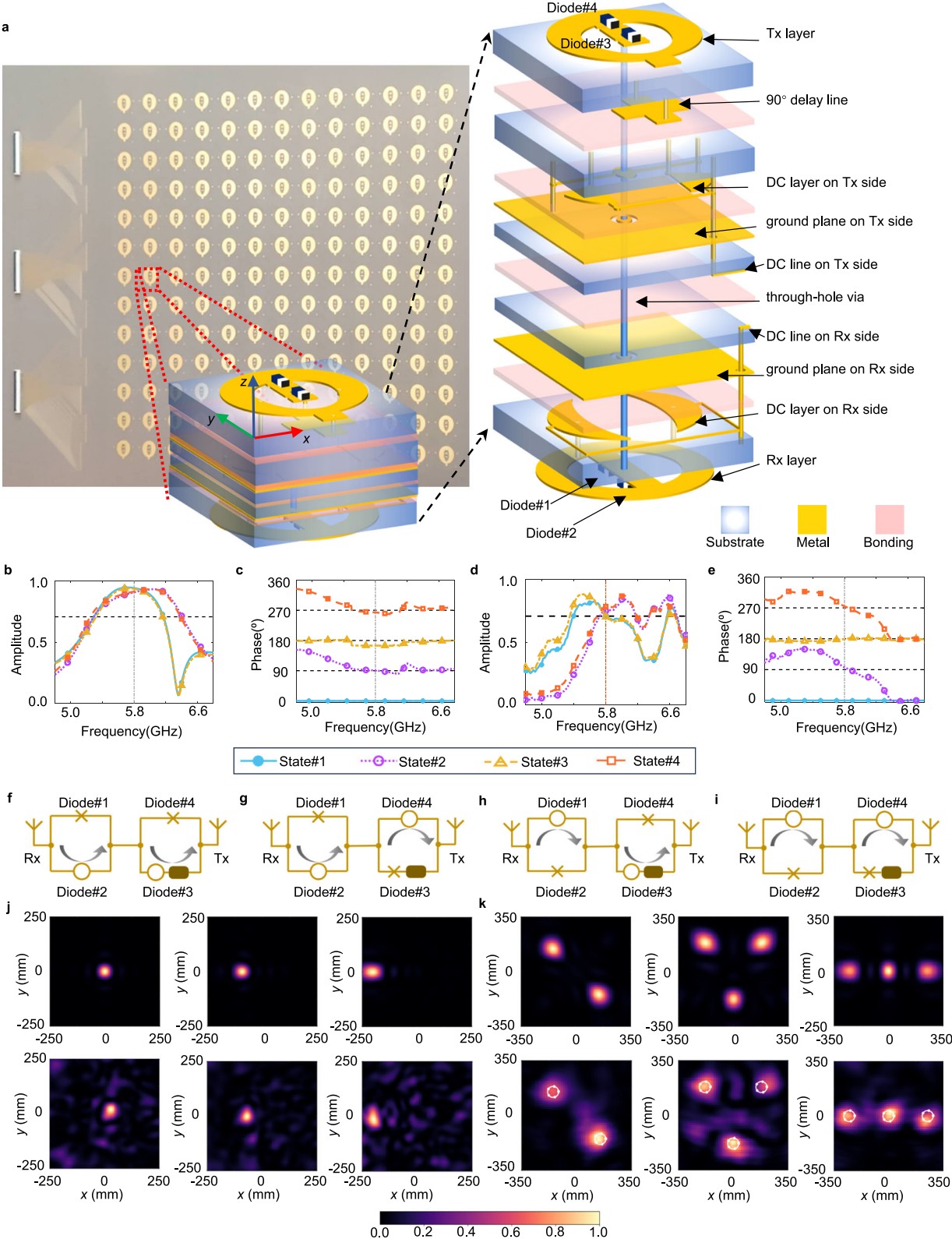

**Fig. 3 | The RTM and its performance analysis. a** The fabricated RTM and its exploded view of the proposed unit-cell. **b**, **c** The simulated transmissive amplitude (**b**) and phase (**c**). **d**, **e** Measured transmissive amplitude (**d**) and phase (**e**). **f**, **g**, **h**, **i** Current paths for State#1 (**f**), State#2 (**g**), State#3 (**h**), and State#4 (**i**). The cross indicates OFF while the circle indicates ON. The rounded rectangle corresponds to the 90° delay line. **j**, **k** The focal spots $|E|^2$ of the reconfigurable power router: (**j**) single focal spot and (**k**) multiple focal spots, where the first row presents the simulated results while the second row presents the measured results. a.u.: arbitrary unit.

flows towards +$y$ axis; when the Diode#1 is ON and Diode#2 is OFF, the current mainly flow towards −$y$ axis. However, this configuration can only provide two-phase levels: 0 and 180°. To achieve two more phase levels, a 90° delay line and another two PIN diodes are added to the transmitter part. As Diode#3 is turned ON and Diode#4 OFF, the transmissive wave gets additional 90° phase shift, enabling 90° and 270° phase levels. Table 1 lists the transmissive phases of the RTM corresponding to the states of the PIN diodes.

The schematics of the current paths for four states are presented in Fig. 3f–i. Comparing State#1 with State#3 (Fig. 3f, h), we can observe that the currents on the unit-cell flow in opposite directions on Rx sides, resulting in a phase difference of 180°. To explain how to achieve an additional 90° phase shift, we take State#1 and State#2 as an example. As presented in Fig. 3f, g, currents flow couterclockwise on Rx sides but in opposite directions on Tx sides, leading to 180° phase difference as discussed above. However, State#2 does not obtain the additional 90° phase shift due to the OFF state of Diode#3 (the delay line is not operative). Therefore, the phase difference between State#1 and State#2 is 180°−90° = 90°. Applying the same analysis to State#3 and State#4, we can induce that the transmissive phase of State#4 is 270°.

The simulated transmissive amplitude and phase responses are shown in the Fig. 3b, c. Here, we define the phase of State#1 as 0 and the phases of other states as the phase differences between them and State#1. Therefore, the phase of State#1 becomes a parallel line over frequency. Although the unit-cell contains four PIN diodes, capable of switch between $2^4$ states, only four states are used. The other states have high reflectivity because the use of PIN diodes is only for changing the current direction on the unit-cell: if two PIN diodes on the same side are simultaneously turned ON or OFF, the current on Rx side flows in opposite directions or is blocked by PIN diodes (also see Supplementary Note 3 for the circuit topology of PIN diodes). From the simulation results shown in Fig. 3b, c, we can observe that the unit-cell is able to switch between four states with respective transmissive phases of 0°, 90°, 180°, and 270°, while maintaining its transmissive amplitude above 0.9 at 5.8 GHz. The transmissive amplitudes are above 0.707 (50% energy efficiency) from 5.32 to 6.16 GHz for four states, indicating a 14.5% bandwidth.

Figure 3d, e shows the measurement results of the fabricated RTM unit-cell. The incident field is $y$ polarized. The experiment setup is shown in Supplementary Fig. S17. The phase responses of the unit-cell at 5.8 GHz for four states are, respectively, 0°, 97°, 179°, and 274°, showing good agreement with the simulated values. However, the amplitude responses are lower compared with those of simulation. The suppression of amplitude responses can be attributed to (1) material loss, including dielectric loss of the substrates and intrinsic resistance of the PIN diodes, and (2) fabrication error. Both factors reduce the amplitude responses. As presented in Supplementary Fig. S11, when we decrease the equivalent resistance of PIN diode, the simulated transmissive amplitude gradually reaches the best performance. To compensate the loss, gain medium may improve the transmissive amplitude[61,62].

We experimentally demonstrate the dynamic beam capability of the proposed framework (see Supplementary Fig. S18 for experimental setup). The reconfigurable power router is formed by cascading the NFF and DBG with a separation distance of 13 mm (about a quarter of working wavelength). We first measured the S-parameter of the router. The results are shown in Fig. 2c. The $S_{11}$ of the NFF are measured both with and without the DBG. We can observe that in both cases the system is resonant at 5.8 GHz. The $S_{11}$ remains resonant (with its magnitude smaller than −10 dB) although it grows larger as the DBG is cascaded. Next, we experimentally show the proposed framework's capability for dynamic beam steering. The experimental setup is presented in Supplementary Fig. S19. The planar nearfield scanning technique is utilized to measure the field distribution with the

**Table 1 | The transmissive phases of the RTM corresponding to the states of the PIN diodes**

| State | Didoe#1 | Diode#2 | Diode#3 | Diode#4 | Phase (°) |
|---|---|---|---|---|---|
| 1 | OFF | ON | ON | OFF | 0 |
| 2 | OFF | ON | OFF | ON | 90 |
| 3 | ON | OFF | ON | OFF | 180 |
| 4 | ON | OFF | OFF | ON | 270 |

scanning area of 700 mm × 700 mm. Figure 3j, k shows the measured results of single focal spot and of multiple focal spots. As can be seen, all focal spots are located near the expected positions. To show that our system is able to manipulate the power distribution over the 3D space, we configured our system to generate two focal spots at two different planes: one spot is located at $z = 300$ mm while the other at $z = 500$ mm, as presented in Supplementary Fig. S22. We can observe that two different spots at (−150, 0, 300) and (200, 0, 500). In Supplementary Fig. S22b, c, the measured results are consistent with the simulated ones. The intensity of the spot at $z = 500$ mm is smaller than that of $z = 300$ mm, which results from the nature of propagation loss as the electromagnetic wave travels. These results reveal good performance of the proposed framework in controlling power distribution at will.

### Environment sensor

In our framework, multiple targets randomly and dynamically appear. We utilize a stereo camera as the environment sensor, cooperating with the intelligent computation unit to capture these dynamic changes. Before deploying the stereo camera, we must calibrate it. The stereo camera calibration consists of single-camera calibration, stereo calibration, and stereo rectification (see Supplementary Note 9 for details). This calibration process retrieves all necessary camera parameters including the camera intrinsic matrix and the distortion coefficients to assist the intelligent computation unit for object detection and localization. The intelligent computation unit consists of a deep-learning-enabled model YOLOv5 for object detection, a coordinate calculator for object localization, and a phase calculator for producing phase patterns for the router. For dark environment, microwave/infrared sensing strategy can help to extract environment information[63,64]. In our framework, optical stereo camera is used as a solution for environment sensing because it's cheap and off-the-shelf.

### Intelligent WPT in dynamic environments

We experimentally demonstrate the capability of our proposed WPT framework to intelligently deliver wireless power to desired objects. Figure 4a shows the experiment setup of the proposed WPT system. The synthesis of phase responses (or the beamforming algorithm) is detailed in Supplementary Note 10. The NFF is connected with a 5.8 GHz signal generator. Three objects (denoted as rectangle, circle, and hole antennas, respectively, see Supplementary Note 6) are connected with three identical rectifying circuits. Some daily electronic devices, such as LED lamp and smartphone, are then connected to the output ports of the rectifying circuits.

In the first experiment, we demonstrated that the proposed WPT system charges static and moving objects selectively, as presented in Fig. 4b, c (also see Supplementary Movies 1 and 2). In this experiment, the input RF power to the router is 8 W. First, the rectangle antenna is placed in a moving car while the circle antenna is placed at the desk. As the moving car passes through the view of the stereo camera, the lamp loaded on the rectangle antenna stays dark. In contrast, the lamp loaded on the circle antenna is lit, as shown in Fig. 4b. Next, under the same configuration, we sweep the objects to be powered: turn on the lamp on the rectangle antenna, but turn off the lamp on the circle antenna. As can be seen from Fig. 4c and Supplementary Movie 2, the

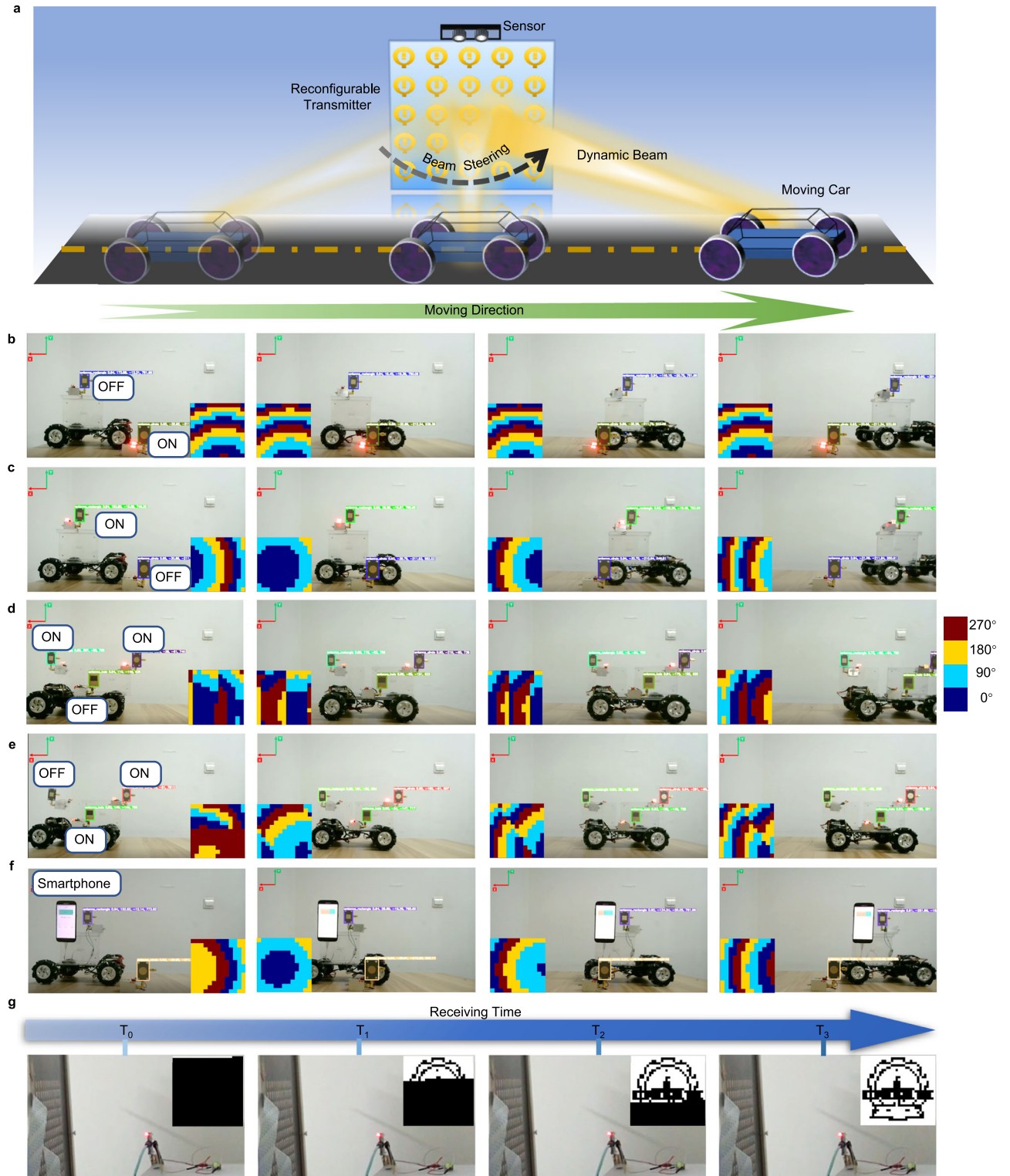

**Fig. 4 | WPT and SWIPT experiments. a** Schematic of the WPT experiment setup. **b** Static and **c** dynamic experimental results of power delivery. **d, e** Dynamic power delivery for three targets: two targets are powered with the third target for reference. **f** A moving smartphone is charged. The insets in (**b**–**f**) are the corresponding phase patterns of the reconfigurable router. **g** Simultaneous power delivery and communication. The insets in (**g**) are the received message.

moving lamp is powered on while the lamp on the desk stays dark. This shows the proposed WPT framework's real-time capability, which allows charging a moving object instantly.

Furthermore, we demonstrated the multiple-target WPT by adding a hole antenna. All three antennas are placed in the moving car as shown in Fig. 4d, e (also see Supplementary Movies 3 and 4). In Fig. 4d and

Supplementary Movie 3, the lumps on the rectangle and circle antennas are lit, while the lump on the hole antenna remains off. Then, we change the lumps to be powered: the lumps on the hole and circle antennas are turned on, but the lump on the rectangle antenna is turned off, as shown in Fig. 4e and Supplementary Movie 4. It is worthy to be noted that the hole antenna is placed on a different plane than the rectangle and circle

antennas. It means that our proposed WPT system is able to focus energy tightly and manipulate power distribution in three-dimensional space, providing significant flexibility for WPT applications.

In the second experiment, a smartphone and a power bank are used as power-consuming targets. The router's input RF power is 50 W in this case. These daily-life electronic devices are connected with the rectangle antenna and placed in the moving car, as shown in Fig. 4f, Supplementary Note 11, and Supplementary Movies 5 and 6. For reference, a circle antenna loaded with a lump is placed on the desk. Both the smartphone and the power bank are charged by the WPT system, while the referenced lump is left off. This experiment demonstrates that the proposed system enables seamless connections with electronic devices in daily life.

Finally, we demonstrate the SWIPT capability of the proposed framework. In this experiment, one port of the circle antenna is connected to an LED for power transfer, while the other port is connected to a spectrum analyzer for wireless communication (see Supplementary Note 13 for details). The spectrum analyzer is used to measure the spectrum of the received signal from the circle antenna. Then, the laptop computer decodes the spectrum and displays the received message on the screen. As presented in Fig. 4g and Supplementary Movie 7, the LED illuminates once the WPT system is powered on. After the frame searching process (finding the head of a frame), the laptop decodes the received signals and displays the corresponding message (the authors' school badge) on the screen. Both wireless information and power transfer are simultaneously realized in our proposed framework.

## Discussion

We have proposed an intelligent WPT framework for multiple dynamic targets. This framework consists of a NFF, a DBG, an off-the-shelf stereo camera, and an intelligent computation unit. The NFF and the DBG, cascaded with a separation of about a quarter wavelength, form a compact reconfigurable power router. The total profile of the NFF and DBG is one order of magnitude smaller than other works. The intelligent computation unit, cooperating with the stereo camera and the deep-learning-driven model YOLOv5, enables simultaneous object detection and localization in real-time. The proposed WPT framework is able to detect the real-time information from dynamic environments and intelligently configure the power distribution, that is, form multiple tight foci in 3D space to selectively deliver wireless power to targets. We also experimentally showed that daily-life devices can be charged seamlessly using the proposed WPT framework. In addition, SWIPT is also demonstrated by using the proposed framework. With the intelligence and flexibility of the WPT framework, we believe that the proposed technology is suitable for a variety of uses, such as low-power sensor networks and industrial power delivery, where sustainable energy supply is in great demand.

Several limitations of the proposed device warrant consideration and optimization for practical applications. The first is the relatively low power transfer efficiency (PTE). It can be attributed to the following reasons: (1) the diffraction-limited spot width; (2) the lossy substrates and diodes; (3) the lossy cables and connectors. As the distance increases, the focal spot increases according to the Abbe diffraction limitation by approximately $R\lambda/D$, where $R$ is the distance between the transmitter (Tx) and the receiver (Rx), $\lambda$ is the wavelength, and $D$ is the Tx size. On the other hand, the fixed Rx antenna size results in the receiver only capturing a partial portion of the delivered power. Diffraction-free beams that can propagate without diffractive spreading, such as Bessel beams, may improve the PTE. The second reason can make the transmissive amplitudes of the RTM unit-cell lower than unity. The advance of material science that enables substrates and diodes with lower losses can alleviate this problem. These cables and connectors are used to connect the router with the signal generator and the receiving antenna with the

rectifier. Although lossless cables and connectors may not be available in practice, shortening or eliminating them is possible by integrating these components (including cables, connectors, the signal generator, et al.) into a chip so as to reduce these losses. The second constraint is electromagnetic interference (EMI). As the delivery power increases, sensitive electronic devices like programmable logic controllers and solid-state drives may be susceptible to EMI. To avoid possible EMI, we can properly design the router's radiation patterns such that the nulls of the radiation patterns point to the directions of sensitive electronic devices. There may also be safety concerns about human health. Integrating a human body sensor into the router can help with this problem. For instance, if the sensor finds human bodies, padding more nulls into radiation patterns, reducing the transmitting power, or closing the router can lessen the influence on human bodies.

The proposed framework has advanced deployments of metasurface-based farfield WPT in terms of low cost, size reduction, selective charging, and SWIPT. Nevertheless, there are still some works left, e.g., to make the framework a reality for practical scenarios. First, the power transfer efficiency can be improved. The best RF-to-RF power transfer efficiency of the proposed work is around 3% (see Supplementary Note 8). It is feasible to deploy the router to deliver power to low-power devices like wearables and RFID, but improving the power transfer efficiency can make the proposed framework applicable to more complex scenarios. Several methods can be used to improve the overall system efficiency, such as better beamforming algorithms to improve the Tx-to-Rx power transfer efficiency, designing metasurfaces with more bits to improve the phase resolution, and better rectifying diodes to improve the RF-to-DC efficiency. An important way to improve the power transfer efficiency is by incorporating multiple receiving antennas[65,66]. This can increase the receiving gain, enabling charging devices with higher power requirements. Second, a farfield WPT standard or protocol is required to meet different practical situations. In unmanned industries, for instance, the maximum permissible power to avoid EMI is required. Besides, it is interesting to connect devices of various power consumption requirements to the router. The router should obtain the power requirements of different devices before delivering power to ensure the quality of service. A practical farfield WPT standard or protocol needs joint efforts from the academic and industrial communities. Finally, the beam angular scan range (BASR) needs to be further improved to fully cover the entire half space. The scanning elevation angle of the proposed router reaches 60° (see Supplementary Note 4). The BASR is mainly limited by the angular dispersion of the metasurface. It means that the unit-cells of the metasurface may have different responses with respect to the incident angles. Metasurface dispersion engineering can reduce the angular dispersion and thus improve the BASR.

Despite these constraints, we can still envision the future applications of the proposed work. The proposed router has been miniaturized to the size of a regular household appliance. In our work, the dimensions of the router are around 0.5 m × 0.5 m × 0.04 m, close to a commercial 27-inch liquid crystal display (LCD). The ability for SWIPT also inspires the integration of multiple systems, such as wireless communication systems, wireless power systems, and wireless sensing systems. We believe that sixth-generation communication, indoor WPT, and SWIPT will benefit from the proposed intelligent WPT router in the future.

## Methods

The experimental setup for nearfield scanning is presented in Supplementary Fig. S19. The device under test is the reconfigurable router, which is formed by cascading the DBG with the NFF, as depicted in Fig. 1. A probe antenna is used as the receiver to detect the amplitude and phase of the spatial electrical field. The probe antenna is moved by

a digitally controlled stage. The movement step is fixed at 10 mm ($\approx$ 0.2$\lambda$). The scanning plane has a size of 700 mm × 700 mm and is located 50 mm away from the reconfigurable router. The amplitude and phase of transmission coefficient (the component $S_{21}$ of the scattering matrix) are recorded by a vector network analyzer (VNA). After the nearfield scanning, we can calculate the field distribution over the focal plane according to the Rayleigh-Sommerfeld diffraction integral[67], which can be expressed as:

$$E(x,y) = \frac{1}{j\lambda} \int_S E(x',y') \frac{z e^{jk\sqrt{(x-x')^2+(y-y')^2+z^2}}}{(x-x')^2+(y-y')^2+z^2} \, \mathrm{d}x'\mathrm{d}y' \qquad (1)$$

where $S$ is the scanning plane, $z$ is the distance between the scanning plane and the focal plane, and $\lambda$ is the operating wavelength, $k = 2\pi/\lambda$.

## Reporting summary

Further information on research design is available in the Nature Portfolio Reporting Summary linked to this article.

## Data availability

All relevant data are available in the paper and its Supplementary Information Files, or from the corresponding author on request. Source data are provided with this paper.

## Code availability

The custom codes utilized in this work are available from the corresponding author on request.

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

## Acknowledgements
This work was partially supported by National Natural Science Foundation of China under Grant no. 62271170 and no. 61731007.

## Author contributions
W.L. and J.Q. conceived the idea and conducted the theoretical analysis. W.L. performed numerical simulations, built the proof-of-principle prototype system, and conducted experiments. W.L., Q.Y., and J.Q. wrote the manuscript. J.H.Q. provided suggestions and comments. J.Q. supervised this work. All authors discussed the results and approved the final manuscript.

## Competing interests
The authors declare no competing interests.
