## [Peer Review File · Nature Communications]

Intelligent Wireless Power Transfer via a 2-Bit Compact Reconfigurable Transmissive-Metasurface-Based RouterREVIEWER COMMENTS

Reviewer #1 (Remarks to the Author):

In this work, the authors propose a compact metasurface-based WPT system that advances the state-of-the-art frameworks in two main aspects: 1) the system design, being compact and low-cost and 2) sensing- and intelligence-optimized system. Overall, the work and results are interesting and top-notch. Nevertheless, in my opinion, the authors could still improve the discussions on the most fundamental concepts. For instance:

1) the authors refer to only a few WPT technologies in the intro, but there are many others. Check, e.g.,
<https://ieeexplore.ieee.org/stamp/stamp.jsp?tp=&arnumber=9319211>

2) the authors discuss metasurfaces for WPT and their advantages but the so-called DMA structures were not discussed. Refer, e.g., to
<https://arxiv.org/abs/2307.01082>
<https://ieeexplore.ieee.org/abstract/document/9738442>
although the latter is not for WPT. Notice that DMAs do not use the reflective property of the metasurfaces but the radiative properties...

3) The statement: "The NFF generates a plane-wave-like wavefront" may be confusing. If the NFF is in the near-field of the DGB, such a wavefront shouldn't be spherical? Maybe the explanation comes later once the construction of the NFF is explained, but at the moment of the statement, it can be very confusing.

Finally, there were no discussions on protocol aspects. For instance, some detected devices may have more energy harvesting requirements than others, and this is not covered by the proposed framework....

Reviewer #2 (Remarks to the Author):

The paper presents the concept of using a reconfigurable periodic structure for wireless power transfer as a router. The topic of wireless power transfer and reconfigurable surfaces is of interest and timely.

Please consider the following comments:

1-In Fig. 2 (i and k) the amplitude is normalized. Is there a reason? How does the maximum amplitude differ in simulated and measured data?

2-The periodic structure is considered to be 13 by 13, why this size is considered?

3-How many diodes are used in total? Please comment on the power consumption of the diodes.

4-More details about the unit-cell simulations should be provided. The unit cell should be characterized for different transmission angles.

5-How unit cell measurement was done? There are large differences between the simulated and measured phase (Fig. 3 (c) and (e)), which that shows the phase bandwidth is much smaller than the simulation. This may be due to the differences between the simulation and measurement setup, but it needs to be commented on.

6- The concept is identical to the Reconfigurable Intelligent Surface concept (RIS). The authors should comment on the differences and similarities of the system with RIS systems. Some references should be added in this regard.

7-Using the optical sensor might be a limiting factor. The authors suggested the use of an Infrared camera. Wouldn't a solution such as a radar or sounding system work better? It is good to have a pros/cons comparison on the choice of the sensor system.

8-I suggest the information about the simulation software be moved before Fig. 2.

- 9-Please comment on Power Transfer Efficiency with and without the reconfigurable surface (such as the experimental examples). Did you measure the power transfer efficiency without the reconfigurable surface? Any comments on the range and angle of beam scanning?
- 10-Please comment on the switching speed. How dynamic a target could be for the system to be able to recognize the location and reconfigure the surface to focus the beam.
- 11-Please explain the details of the beamforming algorithm.
- 12-Please check the manuscript for formatting errors. Please correct "unman" to "unmanned".

Reviewer #3 (Remarks to the Author):

The manuscript presents "Intelligent Wireless Power Transfer via a 2-Bit Compact Reconfigurable Transmissive-Metasurface-Based Router " which is a well-organized and technically sound analysis of the topic at hand. However, the findings in this manuscript do not appear to introduce any significant novelty to the field. To enhance the impact of this work, I suggest the authors consider the following comments.

1. In the introduction, there is not enough information about the studies on self-adaptive metasurfaces and why this research is needed. The research on programmable and self-adaptive metasurfaces have been actively studied. It would be better to provide them as references.
2. Based on my limited knowledge, as for the terminology "SWPIT", it is commonly used as simultaneous wireless information and power transfer rather than simultaneous wireless power and information transfer. I recommend using simultaneous wireless power and information transfer (SWIPT) for consistency.
3. It would be good if you can provide the design procedure of the metasurface unit cell. The reason for choosing that specific design, etc.
4. Design, simulation, and analysis of the rectifier part are missing.
5. Lack of analysis on transfer efficiencies.
Analysis on the Tx to Rx (single and multi-receivers), Rx-load (RF-DC conversion), and entire WPT efficiencies are required.
6. How and why the separation between NFF-DBG is selected to be 13 mm? Normally, a certain separation is required for the metasurface superstrate. It would be great if the authors can provide some scientific evidence on how could the authors reduce the distance between them as it is related to the main contribution (compact size) of this manuscript.

Reviewer #4 (Remarks to the Author):

The paper proposes a compact reconfigurable power router that consists of a plane-wave feeder and a transmissive 2-bit reconfigurable metasurface-based beam generator. The paper is well organized and written with several experiment results. However, there are still concerns that need to be addressed as follows:

1. Please clearly emphasize the novelty of this work compared to the existing works.
2. The authors claimed that the proposed reconfigurable power router has a total thickness of 0.8 wavelength. However, as shown in the global view of the fabricated WPT system in Figure. S9, the overall size of the router with the control board is bigger. For a fair comparison with the existing works, it is recommended to correct the total thickness of the proposed framework since, without the control board, the router could not work.
3. Please thoroughly elaborate on how to design the artificial magnetic conductor (AMC) and partial reflected plane (PRP). How can the authors calculate the dimensions for AMC and PRP? If there is any reference, please cite it.
4. What is the transmitted power in the experiment? Please clarify it.
5. In the first experiment, the authors show that the proposed WPT can light up the LED, which only consumes several mW. However, in the second experiment, they attempted to charge daily-life electronic devices (i.e., a smartphone and a power bank). The power consumption of these devices is relatively high, about 2 to 6 W. Transferring such a high RF power is challenging in WPT system. Please clarify the transmitted power and the received power at these focal spots.
6. What is the power transfer efficiency (PTE) of the system? It would be better to include a figure

- of PTE over the target moving distance to show the effectiveness of the proposed router.
7. In the SWIPT experiment, the authors select space-time modulation rather than PSK or QAM modulation. The data rate is relatively low. Please consider using other modulations.
 8. Grammatical errors and typos should be fixed.

REVIEWER COMMENTS

Reviewer #1 (Remarks to the Author):

In this work, the authors propose a compact metasurface-based WPT system that advances the state-of-the-art frameworks in two main aspects: 1) the system design, being compact and low-cost and 2) sensing- and intelligence-optimized system. Overall, the work and results are interesting and top-notch. Nevertheless, in my opinion, the authors could still improve the discussions on the most fundamental concepts. For instance:

1) the authors refer to only a few WPT technologies in the intro, but there are many others. Check, e.g.,
<https://ieeexplore.ieee.org/stamp/stamp.jsp?tp=&arnumber=9319211>

2) the authors discuss metasurfaces for WPT and their advantages but the so-called DMA structures were not discussed. Refer, e.g., to
<https://arxiv.org/abs/2307.01082>
<https://ieeexplore.ieee.org/abstract/document/9738442>
although the latter is not for WPT. Notice that DMAs do not use the reflective property of the metasurfaces but the radiative properties...

3) The statement: "The NFF generates a plane-wave-like wavefront" may be confusing. If the NFF is in the near-field of the DGB, such a wavefront shouldn't be spherical? Maybe the explanation comes later once the construction of the NFF is explained, but at the moment of the statement, it can be very confusing.

Finally, there were no discussions on protocol aspects. For instance, some detected devices may have more energy harvesting requirements than others, and this is not covered by the proposed framework...

Reviewer #2 (Remarks to the Author):

The paper presents the concept of using a reconfigurable periodic structure for wireless power transfer as a router. The topic of wireless power transfer and reconfigurable surfaces is of interest and timely.

Please consider the following comments:

- 1-In Fig. 2 (i and k) the amplitude is normalized. Is there a reason? How does the maximum amplitude differ in simulated and measured data?
- 2-The periodic structure is considered to be 13 by 13, why this size is considered?
- 3-How many diodes are used in total? Please comment on the power consumption of the diodes.
- 4-More details about the unit-cell simulations should be provided. The unit cell should be

characterized for different transmission angles.

5-How unit cell measurement was done? There are large differences between the simulated and measured phase (Fig. 3 (c) and (e)), which that shows the phase bandwidth is much smaller than the simulation. This may be due to the differences between the simulation and measurement setup, but it needs to be commented on.

6- The concept is identical to the Reconfigurable Intelligent Surface concept (RIS). The authors should comment on the differences and similarities of the system with RIS systems. Some references should be added in this regard.

7-Using the optical sensor might be a limiting factor. The authors suggested the use of an Infrared camera. Wouldn't a solution such as a radar or sounding system work better? It is good to have a pros/cons comparison on the choice of the sensor system.

8-I suggest the information about the simulation software be moved before Fig. 2.

9-Please comment on Power Transfer Efficiency with and without the reconfigurable surface (such as the experimental examples). Did you measure the power transfer efficiency without the reconfigurable surface? Any comments on the range and angle of beam scanning?

10-Please comment on the switching speed. How dynamic a target could be for the system to be able to recognize the location and reconfigure the surface to focus the beam.

11-Please explain the details of the beamforming algorithm.

12-Please check the manuscript for formatting errors. Please correct "unman" to "unmanned".

Reviewer #3 (Remarks to the Author):

The manuscript presents "Intelligent Wireless Power Transfer via a 2-Bit Compact Reconfigurable Transmissive-Metasurface-Based Router " which is a well-organized and technically sound analysis of the topic at hand. However, the findings in this manuscript do not appear to introduce any significant novelty to the field. To enhance the impact of this work, I suggest the authors consider the following comments.

1. In the introduction, there is not enough information about the studies on self-adaptive metasurfaces and why this research is needed. The research on programmable and self-adaptive metasurfaces have been actively studied. It would be better to provide them as references.

2. Based on my limited knowledge, as for the terminology "SWPIT", it is commonly used as simultaneous wireless information and power transfer rather than simultaneous wireless power and information transfer. I recommend using simultaneous wireless power and information transfer (SWIPT) for consistency.

3. It would be good if you can provide the design procedure of the metasurface unit cell. The reason for choosing that specific design, etc.

4. Design, simulation, and analysis of the rectifier part are missing.

5. Lack of analysis on transfer efficiencies.

Analysis on the Tx to Rx (single and multi-receivers), Rx-load (RF-DC conversion), and entire WPT efficiencies are required.

6. How and why the separation between NFF-DBG is selected to be 13 mm? Normally, a

certain separation is required for the metasurface superstrate. It would be great if the authors can provide some scientific evidence on how could the authors reduce the distance between them as it is related to the main contribution (compact size) of this manuscript.

Reviewer #4 (Remarks to the Author):

The paper proposes a compact reconfigurable power router that consists of a plane-wave feeder and a transmissive 2-bit reconfigurable metasurface-based beam generator. The paper is well organized and written with several experiment results. However, there are still concerns that need to be addressed as follows:

1. Please clearly emphasize the novelty of this work compared to the existing works.
2. The authors claimed that the proposed reconfigurable power router has a total thickness of 0.8 wavelength. However, as shown in the global view of the fabricated WPT system in Figure. S9, the overall size of the router with the control board is bigger. For a fair comparison with the existing works, it is recommended to correct the total thickness of the proposed framework since, without the control board, the router could not work.
3. Please thoroughly elaborate on how to design the artificial magnetic conductor (AMC) and partial reflected plane (PRP). How can the authors calculate the dimensions for AMC and PRP? If there is any reference, please cite it.
4. What is the transmitted power in the experiment? Please clarify it.
5. In the first experiment, the authors show that the proposed WPT can light up the LED, which only consumes several mW. However, in the second experiment, they attempted to charge daily-life electronic devices (i.e., a smartphone and a power bank). The power consumption of these devices is relatively high, about 2 to 6 W. Transferring such a high RF power is challenging in WPT system. Please clarify the transmitted power and the received power at these focal spots.
6. What is the power transfer efficiency (PTE) of the system? It would be better to include a figure of PTE over the target moving distance to show the effectiveness of the proposed router.
7. In the SWIPT experiment, the authors select space-time modulation rather than PSK or QAM modulation. The data rate is relatively low. Please consider using other modulations.
8. Grammatical errors and typos should be fixed.

Response to the Reviewers' Comments: We would like to express our gratitude to the editor and reviewers for appreciation and constructive comments on our manuscript. We are also thankful for their inspiring comments which have allowed us to greatly improve the manuscript. We have followed the reviewers' advices and carefully modified the manuscript. Hopefully, our reply and the corresponding revision made to the manuscript can meet all requirements of the reviewers. Our detailed replies are listed as follows.

Responses to Reviewer #1:

Comment: Reviewer #1 (Remarks to the Author):

In this work, the authors propose a compact metasurface-based WPT system that advances the state-of-the-art frameworks in two main aspects: 1) the system design, being compact and low-cost and 2) sensing- and intelligence-optimized system. Overall, the work and results are interesting and top-notch. Nevertheless, in my opinion, the authors could still improve the discussions on the most fundamental concepts. For instance:

Response: We thank the reviewer for the positive comments, which encouraged us a lot. The insightful comments are very constructive for further improvement of this work. In the following, we address the specific comments point-by-point whilst revising our manuscript.

Comment: 1) the authors refer to only a few WPT technologies in the intro, but there are many others. Check, e.g.,

<https://ieeexplore.ieee.org/stamp/stamp.jsp?tp=&arnumber=9319211>

Response: We thank the reviewer for this valuable comment. Following the advice of the reviewer, we have supplemented other WPT technologies in our revised manuscript for more complete discussion on current WPT development.

Added or Revised Text:

In the main text:

"Ambient and dedicated WPT technologies are developed to accelerate the growth of IoT [8][9]. The ambient WPT enables capturing wireless energy from environments, such as sunlight, wind, thermal energy, and wireless microwave energy. The availability of the ambient energy usually relies on the time, location, weather, and so on. On the other hand, the dedicated WPT is characterized by delivering wireless energy through the energy network infrastructure, providing a steady energy supply strategy."

Comment: 2) the authors discuss metasurfaces for WPT and their advantages but the so-called DMA structures were not discussed. Refer, e.g., to

<https://arxiv.org/abs/2307.01082>

<https://ieeexplore.ieee.org/abstract/document/9738442>

although the latter is not for WPT. Notice that DMAs do not use the reflective property of the metasurfaces but the radiative properties...

Response: We thank the reviewer for this significant comment. The DMA structures are also an important candidate for WPT. We have supplemented the discussion about DMA and the corresponding references in our revised manuscript.

Revised Text:

In the main text:

"Dynamic metasurface antennas (DMA), one of the metasurface-based WPT technologies, consist of a series of reconfigurable-unit-cells-loaded waveguides connected to a set of radio-frequency (RF) chains [33][34][35]."

Comment: 3) The statement: "The NFF generates a plane-wave-like wavefront" may be confusing. If the NFF is in the near-field of the DGB, such a wavefront shouldn't be spherical? Maybe the explanation comes later once the construction of the NFF is explained, but at the moment of the statement, it can be very confusing.

Response: We thank the reviewer for this valuable comment. We would like to explain why we designed the NFF. Conventional feeders for metasurface-based WPT systems, such as horn antennas, usually radiate a non-uniform wavefront, which, as the reviewer points out, is typically spherical in the near-field. They are required to be placed several to ten wavelengths or more away from a metasurface so as to acquire a uniform wavefront (that is, to enable properly exciting all unit-cells on the metasurface). This point is the reason why the use of the conventional feeders results in large system profiles. To reduce the profile, we design a low-profile NFF, which is able to generate a uniform wavefront in the nearfield. We would like to give a qualitative picture to further explain the working mechanism of the NFF. The proposed NFF are composed

of a 2×2 antenna array, an artificial magnetic conductor (AMC) and a partial reflective plane (PRP). The latter two components create a Fabry-Perot cavity (FPC). The powers are injected into the FPC through the antenna array. Due to the highly reflective properties of the AMC and the PRP, most of the injected powers first spread through the FPC (going back and forth between the AMC and the PRP). Arriving at the PRP, the injected powers partially leak out due to the partial reflection property of the PRP. As this process repeats, a uniform wavefront is formed in the nearfield of the NFF, as shown in Figure. R1 - 1. Such a nearfield distribution can be directly fed to the metasurface to acquire uniform excitation.

To avoid possible confusion, we have revised the statement about the NFF to make the manuscript more readable.

Figure. R1 - 1. The qualitative explanation of the NFF. From left to right, the first two layers are AMC and PRP, respectively, the third layer is the metasurface.

Revise Text:

In the main text:

"Different from conventional feeders that generate a non-uniform wavefront in the nearfield, the NFF is designed to generate a uniform wavefront (a plane-wave-like wavefront in the nearfield, as shown in the following section) so as to properly excite the entire DBG."

Comment: Finally, there were no discussions on protocol aspects. For instance, some detected devices may have more energy harvesting requirements than others, and this

is not covered by the proposed framework....

Response: We thank the reviewer for the insightful comment. The WPT protocol is very important for applications in practical scenarios. As the reviewer points out, different devices may require different energy harvesting requirements. Practically, it is necessary for the proposed router to determine how much power should be transferred to the receivers. For short ranges, wireless power supply (WPS) by inductive WPT has been commercialized by some products and standards (such as Qi, PMA, and Rezence). However, farfield WPT protocols or standards are still an open research topic. We try to provide a possible farfield WPT protocol and have some discussions on it. Please see the revised Supplementary Note 12.

Revised Text:

In the Supplementary Note:

"For IoT scenarios, there are many kinds of devices. They may have different energy-harvesting requirements. Although several standards for inductive WPT, such as Qi, PMA, and Rezence, have been released, there is no widely accepted farfield WPT protocol or standard. Therefore, for a general WPT router, a farfield WPT protocol is required for the compatibility of different devices. Here, we would like to provide a possible protocol for farfield WPT.

As shown in Figure. S32, the process of establishing WPT between one transmitter and one or multiple receivers has three phases and is described as follows:

1. **Detection phase.** The transmitter detects the receiver appearing within the operating range, records the receiver's position, and then sends a ping signal to the receiver. The receiver, if passive, is powered by the ping signal and then sends a control tag back to the transmitter. The control tag should contain the receiver ID and maximum power capacity (MPC). The receiver ID is used to identify the receiver since multiple receivers may appear. The MPC is used to indicate how much power the receiver needs (if MPC=0, the receiver has no energy requirement). The transmitter will deliver power to the receiver(s) according to the control tag.

2. **Identification phase.** The transmitter records the tag and associates it with the corresponding receiver's position. The receiver with energy requirements should be ready to receive wireless power. If there are multiple receivers, the detection and identification phases are performed for each receiver in sequence.
3. **Power transfer phase.** The transmitter delivers power to the receiver according to the receiver ID and MPC. The receiver not only receives the delivery power but also sends the control tag periodically until the power requirement is satisfied (for example, if the receiver is fully charged, it sends a tag with MPC=0 to the transmitter).

Figure. S32. The flow chart for the WPT protocol

In the detection phase, we assume the receiver is passive. It is not a necessary condition. For an active receiver, the tag can be sent to the transmitter once the ping signal is received. In the above protocol, the messages sent between the transmitter and the receiver are small amounts of data including the device MPC and ID. Therefore, establishing WPT between them takes a short time. Besides, multiple beams are formed simultaneously. Each beam has a corresponding intensity for each device's MPC. In this

way, multiple devices are charged simultaneously with their corresponding energy requirements.

Indeed, this protocol is relatively raw. For practical applications, there are many issues that need to be considered, such as the operating range, operating frequency, receiving antenna size and type, and so on. Therefore, a practical farfield WPT protocol (or standard) is now an open research topic and needs joint efforts from both academic and industrial communities."

Responses to Reviewer #2:

Comment: Reviewer #2 (Remarks to the Author):

The paper presents the concept of using a reconfigurable periodic structure for wireless power transfer as a router. The topic of wireless power transfer and reconfigurable surfaces is of interest and timely.

Please consider the following comments:

Response: We appreciate the reviewer's positive comments, which gave us a lot of encouragement. The insightful comments are extremely helpful in further refining our manuscript. In the following, we address the specific comments point-by-point whilst revising our manuscript.

Comment: 1-In Fig. 2 (i and k) the amplitude is normalized. Is there a reason? How does the maximum amplitude differ in simulated and measured data?

Response: We thank the reviewer for this comment. As the reviewer points out, the amplitude is normalized. We normalize the simulated and measured amplitudes for a clearer comparison, eliminating the influence of input power and, in turn, making the simulated and measured values comparable. In the experiments, the measurement system contains many microwave connectors and coaxial cables, which cause losses in the input power. Eliminating all the losses is difficult. The normalization helps us better compare the simulated and measured amplitudes without being affected by the input power.

The difference in the maximum amplitudes between the simulated and measured data can be attributed to the following reasons:

- 1) The intrinsic loss of the substrates used to fabricate the nearfield feeder (NFF).
- 2) Installation error of the NFF. Practically, it is difficult to keep the two components of NFF (that is, the artificial magnetic conductor (AMC) and the partial reflective plane (PRP)) strictly parallel to each other.

3) Fabrication error. Limited by the practical fabrication process, the fabricated AMC and PRP are not 'ideally flat surfaces' but 'curved surfaces', as shown in Figure. R2 - 1. This introduces phase noise as the input power bounces between the AMC and PRP, resulting in the deviation between the simulated and measured field distribution.

Figure. R2 - 1. (a) The ideal AMC and PRP are 'flat surfaces'. (b) The fabricated AMC and PRP are 'curved surfaces' due to the imperfect fabrication.

Comment: 2-The periodic structure is considered to be 13 by 13, why this size is considered?

Response: We thank the reviewer for this comment. The size of the metasurface is set according to the following reasons: 1) Metasurfaces are periodic structures that require enough unit-cells to satisfy the periodicity; otherwise, they cannot effectively manipulate electromagnetic waves. 2) From an industrial design point of view, if the metasurface is too large, it may not be well accepted by the commercial market, especially indoor wireless charging scenarios. A common household appliance usually has a size smaller than 0.5–1 m. Larger sizes may limit the commercial acceptance. The cross section of the proposed WPT system is around 0.5 m × 0.5 m, while the metasurface has a size of 0.3 m × 0.3 m. Some space is reserved for assembling the WPT system or future extensions. Therefore, the size of the metasurface is a good choice to balance among the above reasons.

Comment: 3-How many diodes are used in total? Please comment on the power consumption of the diodes.

Response: We thank the reviewer for this comment. The proposed metasurface consists

of 13×13 unit-cells, each of which contains 4 diodes. The total diodes are $13 \times 13 \times 4 = 676$. For all phase states $\{0^\circ, 90^\circ, 180^\circ, 270^\circ\}$, two diodes are ON while the other two are OFF on each unit-cell (each unit-cell has four diodes). In the experiments, the applied voltage to each diode is 0.8 V. The diode current is 2.4 mA. Therefore, the power consumption of all diodes is $676/2 \times 0.8 \times 0.0024 = 0.65$ W.

We would like to explain why the applied voltage is set to 0.8 V. Figure. R2 - 2 shows the simulated transmissive amplitudes under different applied voltages. As can be seen, the transmissive amplitudes increase for all phase states as the applied voltage increases. It is due to the fact that as the applied voltage increases (0.7 V \rightarrow 0.8 V \rightarrow 0.9 V), the equivalent resistance of the diode decreases ($5 \Omega \rightarrow 1.3 \Omega \rightarrow 0.6 \Omega$). The diode current also increases (0.23mA \rightarrow 2.4 mA \rightarrow 49 mA). We can see that the resistance trends toward saturation as the applied voltage increases from 0.8 V to 0.9 V. For each applied voltage, the average amplitudes of four states can be calculated as follows:

$$(1) 0.7 \text{ V: } (0.87 + 0.84 + 0.86 + 0.85) / 4 = 0.86$$

$$(2) 0.8 \text{ V: } (0.96 + 0.92 + 0.96 + 0.93) / 4 = 0.94$$

$$\text{relative increase: } (0.94 - 0.86) / 0.86 = 9.30\%$$

$$(3) 0.9 \text{ V: } (0.98 + 0.94 + 0.97 + 0.95) / 4 = 0.96$$

$$\text{relative increase: } (0.96 - 0.86) / 0.86 = 11.63\%.$$

The relative increases are 9.30% from 0.7V to 0.8V and 11.63% from 0.7V to 0.9V. The power consumption with an applied voltage of 0.7V is $676/2 \times 0.7 \times 0.00023 = 0.055$ W. The power consumption with an applied voltage of 0.8 V is 11.82 times greater than that with an applied voltage of 0.7 V. However, the power consumption with an applied voltage of 0.9 V increases up to $676/2 \times 0.9 \times 0.049 = 14.91$ W, which is 271.09 times greater than that with an applied voltage of 0.7 V. Such an increase in power consumption only brings about a 2.33% increase in the transmissive amplitude. Therefore, the choice of the applied voltage is a balance between the increase in the transmissive amplitude and the increase in the power consumption.

Figure. R2 - 2. Simulated transmissive amplitudes vs different voltages. (a) State#1: 0°; (b) State#2: 90°; (c) State#3: 180°; (d) State#4: 270°.

Comment: 4-More details about the unit-cell simulations should be provided. The unit cell should be characterized for different transmission angles.

Response: We thank the reviewer for this important comment. We have supplemented the unit-cell simulation setups, which include the boundary condition, the port setting, the diode model, et al. We have also provided the characterization of the unit-cell for different transmission angles. Please see the revised Supplementary Note 2 and 3.

Revised Text:

i) In the Supplementary Note 2:

"(1) Simulation Setup for the AMC and PRP unit-cells

We performed the electromagnetic simulation of the unit-cell by using the commercial software CST Microwave Studio. Figure. S3 shows the modeled AMC and PRP unit-cells. The Unit Cell boundary condition is applied in the $\pm x$ and $\pm y$ directions. Port#1 and Port#2 are set in the $-z$ and $+z$ directions, respectively. Both ports are specified as the Floquet Port. The polarization direction of incident waves is set in the y direction.

Figure. S3. Simulation setup. (a) AMC and (b) PRP unit-cells"

ii) In the Supplementary Note 3:

"(2) Simulation Setup for the RTM Unit-Cell

The electromagnetic simulation setup of the RTM unit-cell is also carried out by the commercial software CST Microwave Studio. Figure. S9(a) shows the modeled unit-cell with a surrounding boundary box. The simulation configuration is the same as the simulations for the AMC and PRP unit-cells. The PIN diode is SMP1320-040LF from SKYWORKS. Its equivalent circuit model in the RTM simulation setup is shown Figure. S9(b). For OFF state, the diode is equivalent to a series RLC circuit. For ON state, the diode is equivalent to a resistor. The incident polarization is set in the y direction.

Figure. S9. The simulation setup for the RTM unit-cell. (a) 3D view of the simulation model. (b)The equivalent circuit model of the PIN diode."

iii) In the Supplementary Note 3:

(3) The Characterization of the Proposed Unit-Cell Under Different Incident Angles

Figure. S12 shows the simulated transmissive amplitude and phase responses of the propose RTM unit-cell. It can be seen that the transmissive amplitudes decrease as the incident angle increases. The bandwidths of the transmissive phases also decrease. Nonetheless, the transmissive phases at 5.8GHz remain relatively stable, that is, stay around 0° , 90° , 180° , and 270° for the corresponding states, showing robust phase responses with respect to the incident angle.

Figure. S12. The simulated transmissive amplitudes and phases under different incident angles: (a) 15° ; (a) 30° ; (a) 45° ; (a) 60° . "

Comment: 5-How unit cell measurement was done? There are large differences between the simulated and measured phase (Fig. 3 (c) and (e)), which that shows the phase bandwidth is much smaller than the simulation. This may be due to the differences between the simulation and measurement setup, but it needs to be commented on.

Response: We thank the reviewer for this comment. The experimental setup for transmissive amplitude and phase measurements is shown in Figure. R2 - 3. Two horn antennas are placed on both sides of the fabricated metasurface, respectively, and connected to a vector network analyzer (VNA).

Firstly, we calibrated this experimental setup by measuring the transmission (S_{21}) between two horn antennas without the metasurface. Next, the metasurface is placed between the two horn antennas, and the corresponding transmissions of four states are measured. Finally, we can calculate the transmissive coefficients of the metasurface as follows:

$$T^1 = \frac{S_{21}^1}{S_{21}^c}, \quad T^2 = \frac{S_{21}^2}{S_{21}^c}, \quad T^3 = \frac{S_{21}^3}{S_{21}^c}, \quad T^4 = \frac{S_{21}^4}{S_{21}^c}$$

where T indicates the transmissive coefficient, and the superscripts $\{1, 2, 3, 4, c\}$ indicate State#1, State#2, State#3, State#4, and the calibrated state (no metasurface is placed between the two horn antennas), respectively. The transmissive amplitudes and phases of the metasurface are the moduli and angles of the transmissive coefficients, respectively.

Figure. R2 - 3. The experimental setup for the unit-cell measurement.

The difference between the simulated and measured phases can be attributed to the following reasons:

1) As the reviewer points out, the simulation and measurement setups are not the same. The unit-cell is illuminated by plane waves in the simulation but by quasi-spherical waves in the measurement. In the simulation, the plane waves are used to satisfy the periodic boundary condition. To obtain plane-wave illumination in the measurement, the horn antenna is required to be placed in the farfield region. However, the farfield distance is much greater than the size of the metasurface, resulting in a

received field that is almost the diffraction field. Therefore, both horn antennas are placed in the nearfield to perform the measurement so as to eliminate the effect of the diffraction field on the measurement results. Due to the placement of the horn antennas in the nearfield, the illumination is quasi-spherical, which contributes to the difference between the simulation and measurement setups.

2) The capacitance values of diodes are inconsistent: there is a discrepancy between the actual capacitance value and the capacitance value in the data sheet, and each diode cannot be completely the same. The control board cannot guarantee that each output port is strictly 0.8 V due to the output voltage accuracy of the chips and the resistance error of the chip resistors. The limited accuracy of the output ports makes the states of the diodes not strictly equivalent to each other. As a result of these inconsistencies, the parameters of the practical diode are not exactly the same as the simulated ones.

3) Fabrication error. Due to the multi-layer property, the metasurface needs to be pressed multiple times in fabrication processing. The desired thickness of the metasurface is 5.8 mm. Due to the fabrication error, the thickness of the fabricated metasurface is around 5.3 mm. Besides, the fabricated metasurface is also not the ideal 'flat surface' assumed in the simulation. These fabrication errors may also contribute to the difference between the simulated and measured amplitudes.

Comments: 6- The concept is identical to the Reconfigurable Intelligent Surface concept (RIS). The authors should comment on the differences and similarities of the system with RIS systems. Some references should be added in this regard.

Response: We thank the reviewer for this comment. RIS is also referred to as intelligent reflecting surface [R1], large intelligent surface [R2], et al. They are included in a more general topic: reconfigurable metasurfaces/metamaterials. The use of reconfigurable metasurfaces/metamaterials ranges from microwave to optics. In microwave, the reconfigurability is usually realized by loading diodes, varactors, transistors, et al. In optics, it is realized by using graphene, phase-change material, et al. In our manuscript, we use the general terminology 'reconfigurable metasurface' for broader readers.

RIS is considered an excellent candidate technology for sixth-generation (6G) communication. It is aimed at searching energy- and spectral-efficient solutions for wireless communication systems. RIS is also envisioned to create smart and reconfigurable wireless channels by overcoming the randomness of natural environments.

Although both our proposed system and RIS systems contain reconfigurable metasurfaces, our manuscript focuses on providing an intelligent wireless power transfer (WPT) solution. Compared with current metasurface-based WPT systems, which are usually reflective and have large profiles, our proposed system enables 2-bit phase manipulation on the transmissive waves and processes a subwavelength profile. Multi-target WPT is realized by our proposed system. In addition, the multi-target WPT is selective; that is, it powers some targets while leaving other targets unpowered, even when all targets are moving. The proposed system also allows simultaneous wireless information and power transfer (SWIPT), as we have experimentally demonstrated in the main text.

The reviewer's comment gives us insightful inspiration about the proposed system. We would like to express our gratitude to the reviewer for leading us to think about transplanting the proposed system to the RIS wireless communication field.

[R1] Wu Q, Zhang S, Zheng B, et al. Intelligent reflecting surface-aided wireless communications: A tutorial[J]. IEEE Transactions on Communications, 2021, 69(5): 3313-3351.

[R2] Hu S, Rusek F, Edfors O. Beyond massive MIMO: The potential of data transmission with large intelligent surfaces[J]. IEEE Transactions on Signal Processing, 2018, 66(10): 2746-2758.

Revised Text:

In the main text:

"In the quest for energy- and spectral-efficient solutions for wireless communication systems, reconfigurable intelligent surfaces (RIS) have emerged as a vital area of

exploration [54][55][56][57][58][59]. They are envisioned to create reconfigurable wireless environments for the next generation of wireless communication."

Comment: 7-Using the optical sensor might be a limiting factor. The authors suggested the use of an Infrared camera. Wouldn't a solution such as a radar or sounding system work better? It is good to have a pros/cons comparison on the choice of the sensor system.

Response: We thank the reviewer for this comment. It is feasible to use other sensor systems, as the reviewer points out, like radar or sounding systems. The use of optical cameras is based on the following reasons:

- 1) An optical camera is a mature commercial product. It is low-cost and off-the-shelf.
- 2) An optical sensor can be regarded as a 'passive' sensor. It uses environmental light without the need for external illumination sources. In indoor environments, for example, an optical sensor can naturally use the ambient light, such as sunlight and lamplight.
- 3) An optical camera has a good resolution, which is convenient for object detection and localization. We will explain this in the following points.
- 4) The object detection model we used is YOLOv5, which is usually used to detect optical images. In addition, most of the public datasets currently used for object detection are optical, and thus using an optical camera helps us use these public datasets to accelerate the training and debugging of the object detection model.
- 5) In our manuscript, the 3D localization principle is triangulation. To achieve such localization, only a binocular camera (composed of two independent, small, and inexpensive optical cameras) is needed. In addition, the coordinate estimation based on triangulation requires less computation since the solution is analytical, which helps to achieve fast object localization and tracking.

Indeed, the optical sensor is limited by dark environments. In that case, it is necessary to replace optical sensors with sensors in other bands, such as infrared, microwave, or sounding sensors. In some possible foggy weather, microwave or sounding sensors could be better. Adopting integrated sensing and communication (ISAC) technologies,

microwave sensors may be integrated into our proposed system, which could be a future work for the authors. Due to the penetrability, microwaves can also detect objects under optical non-transparency. However, compared with optical sensors, microwave/sounding sensors often have relatively larger volumes, higher costs, and relatively complex positioning algorithms. Microwave sensors usually need to emit a probing signal, which may result in multipath effects in indoor environments. It is intractable to deal with the multipath effects. We have made a pros-and-cons comparison on the choice of the sensor system. Please see Table R2-1.

Table R2-1. The pros and cons of three types of sensors.

type \ term	size	resolution	cost	penetrability	light sensitivity
Optical	small	high	low	low	high
Microwave	large	medium	high	high	low
Sounding	large	medium	high	high	low

Comment: 8-I suggest the information about the simulation software be moved before Fig. 2.

Response: We thank the reviewer for this comment. The information about the simulation software has been moved before Fig.2 in the revised manuscript.

Revised Text:

In the main text:

"We performed the electromagnetic simulation using the commercial software CST Microwave Studio."

Comment: 9-Please comment on Power Transfer Efficiency with and without the reconfigurable surface (such as the experimental examples). Did you measure the power transfer efficiency without the reconfigurable surface? Any comments on the range and angle of beam scanning?

Response: We thank the reviewer for this comment. We have measured the power transfer efficiency with and without the reconfigurable metasurface. Please see Figure S32 in the revised Supplementary Note 8. We have also supplemented analysis on the power transfer efficiency with multiple receivers. Please see Figure S33 and S34 in the revised Supplementary Note 8. The beam scanning range is supplemented in the revised Supplementary Note 4. The beam scanning range is from -60° to 60° , as shown in Figure. S19 and S20. When the scanning angle θ_0 increases, the sidelobe level gradually increases. For $\theta_0 = 70^\circ$, it is difficult to distinguish from the mainlobe, as shown in Figure. S20(h). The deviation between the theoretical and measured values mainly results from the angular dispersion of the metasurface unit-cells. As shown in Figure. S12, the metasurface unit-cell's performance drops with the increase of the incident angle. The angular dispersion is not considered in the theoretical calculations, that is, in Equation (3) of Supplementary Note 4.

Revised Text:

In the Supplementary Note 4:

"(5) Single- and Multiple-Beam Scanning

According to the antenna array theory, the theoretical radiation pattern of an array can be represented as

$$F(\theta, \varphi) = \sum_m \sum_n A_{m,n} \exp \left\{ jk \left[(m-1)d_x \sin \theta \cos \varphi + (n-1)d_y \sin \theta \sin \varphi \right] \right\} \quad (3)$$

where (m, n) indicates the m th row and n th column element, $A_{m,n}$ is the complex amplitude of the (m, n) th element, k is the wave number, d_x (d_y) is the element period along x (y), and θ and φ are the elevation and azimuth angles, respectively. In the case of the proposed 2-bit router, the complex amplitude $A_{m,n}$ can take on values $\{e^{j0}, e^{j\pi/2}, e^{j\pi}, e^{j3\pi/2}\}$.

Figure. S19(a) shows the measured scanning beams of the proposed router. Its scanning elevation angle reaches 60° . Figure. S20 shows the comparison between the theoretical and measured results. As can be seen, with the increase of the scanning angle θ_0 , the sidelobe level gradually approaches the mainlobe level. For $\theta_0 = 70^\circ$, it is difficult to distinguish from the mainlobe. We also conducted multiple-beam scanning

experiments. The measured multiple-beam results are shown in Figure. S21. For the single- and multiple-beam results, the deviation between the theoretical and measured values mainly results from the angular dispersion of the metasurface unit-cells. As shown in Figure. S12, the metasurface unit-cell's performance drops as the incident angle increases. The angular dispersion is not considered in the theoretical calculations, that is, in Equation (3). The metasurface dispersion engineering can help with this issue.

Figure. S19. (a) The measured beam scanning with the scanning angle θ_0 ranges from 0° to 60° . (b) The theoretical and measured θ_{HPAR} .

The scan resolution of the proposed system can be estimated by its half-power angle resolution (HPAR). The HPAR is a widely accepted criticism and is also used as resolution estimation in radars, telescopes, and microscopes. The proposed router's HPAR can be estimated as

$$\theta_{\text{HPAR}} = \frac{\lambda}{D} \quad (4)$$

where λ is the working wavelength and D is the effective aperture size of the metasurface. In our case, the working wavelength is 51.7 mm, and the size of the metasurface is 300 mm. The HPAR of the proposed system can be estimated as $\theta_{\text{HPAR}} = 51.7/300 = 9.87^\circ$. With the scanning angle $\theta_0 = 0^\circ$, the measured HPAR is 11.36° , which is close to the theoretical value. As shown in Figure. S19(b), the trend of the measured HPAR is similar to the theoretical one. As the effective aperture size decreases with the increase of the scanning angle, the measured HPAR becomes lower (that is, the beamwidth increases). The angular dispersion of the metasurface can contribute to the deviation between the theoretical and measured results.

Figure. S20. The comparison between the theoretical and measured results: (a) $\theta_0 = 0^\circ$; (b) $\theta_0 = 10^\circ$; (c) $\theta_0 = 20^\circ$; (d) $\theta_0 = 30^\circ$; (e) $\theta_0 = 40^\circ$; (f) $\theta_0 = 50^\circ$; (g) $\theta_0 = 60^\circ$; (h) $\theta_0 = 70^\circ$.

Figure. S21. The measured multiple-beam scanning. (a) – (c) are double-beam patterns. (d) – (f) are triple-beam patterns. (a) $(\theta_1, \theta_2) = (-30^\circ, 20^\circ)$; (b) $(\theta_1, \theta_2) = (-45^\circ, 45^\circ)$; (c) $(\theta_1, \theta_2) = (10^\circ, 40^\circ)$; (d) $(\theta_1, \theta_2, \theta_3) = (-45^\circ, 0, 45^\circ)$; (e) $(\theta_1, \theta_2, \theta_3) = (-30^\circ, 20, 40^\circ)$; (f) $(\theta_1, \theta_2, \theta_3) = (-50^\circ, 10^\circ, 30^\circ)$.

"

In the Supplementary Note 8:

" Supplementary Note 8: WPT efficiencies

The entire WPT system consists of a commercial tunable signal generator, the proposed router, a receiving antenna, and a rectifier, as illustrated in Figure. S31. To fully characterize the entire system, the DC-to-DC efficiency of all the components, from the DC input to the DC output, can be considered as follows:

$$\eta_{DD} = \eta_{DR} \eta_{RR} \eta_{RD} \quad (4)$$

where η_{DR} is the efficiency of the commercial tunable signal generator (or DC-to-RF efficiency), η_{RR} is the RF-to-RF efficiency, and η_{RD} is the rectifier efficiency (or RF-to-

DC efficiency). In our proposed system, η_{DR} has a typical value of 40%. The rectifier efficiency η_{RD} has been shown in Figure. S30. In the following, we focus on the RF-to-RF efficiency and the DC-to-DC efficiency.

Figure. S31. The efficiency of the WPT system.

(1) RF-to-RF Efficiency

The experimental setup is shown in Figure. S31. The antennas shown in Supplementary Note 6 are used as the receiving antennas in this section. We first measured the RF-to-RF efficiency between the NFF and the receiving antenna (denote as the NFF group). Next, we inserted the DBG to constitute the router and measured the corresponding RF-to-RF efficiency (denote as the NFF+DBG group). To illustrate the improvement after the insertion of the DBG, we define the beam gain as

$$G = 10 \log_{10} \left(\frac{P_{ND}}{P_N} \right) \quad (5)$$

where P_{ND} is the received power of the NFF+DBG group and P_N is the received power of the NFF group. The corresponding results are shown in Figure. S32.

(1.1) Single-Target RF-to-RF Efficiency

Figure. S32 shows the single-target RF-to-RF efficiency. In this case, only a single receiving antenna is used. As can be seen from Figure. S32(a) and (b), as the distance d increases, the RF-to-RF efficiency decreases. This is due to the spreading nature of electromagnetic waves. From Figure. S32(a), for $\theta = 0$, the RF-to-RF efficiency is lowered after the insertion of the DBG. This results from the transmissive amplitude of the RTM being not unity, as shown in Figure 3(d) and (e) in the main text. As shown in Figure. S32(c), the average gain (loss) is -3.72 (3.72) dB, which is close to the measured

transmissive amplitudes of the RTM shown in Figure 3(d) and (e) in the main text. In contrast, for $\theta = 45^\circ$, the RF-to-RF efficiency is improved after the insertion of the DBG, as shown in Figure. S32(b). The average gain is 13.41 dB. The insertion of the DBG enables the reconfigurable beams to redirect the transmitted powers to the receiving antenna and, in turn, improves the RF-to-RF efficiency while the receiver is placed at off-axis positions.

Figure. S32. The measured single-target RF-to-RF efficiency: (a) $\theta = 0^\circ$; (b) $\theta = 45^\circ$. (c) Gain after inserting the DBG. NFF represents the nearfield feeder, and DBG represents the dynamic beam generator.

(1.2) Multiple-Target RF-to-RF Efficiency

We also measured the multiple-target RF-to-RF efficiency. In the first case, the experimental setup with two receiving antennas (denoted as R#1 and R#2, respectively) is considered. The two receiving antennas are placed at -45° and $+45^\circ$, respectively, as shown in Figure. S33(a). The corresponding measured RF-to-RF efficiencies are shown in Figure. S33(b). We can see that the total RF-to-RF efficiency (R#1+R#2) in Figure. S33(b) is close to that of single target with $\theta = 45^\circ$ in Figure. S32(b). This indicates that the transmitted power is properly split into two parts, one for R#1 and the other for R#2. In the second case, three receiving antennas (denoted as R#1, R#2, and R#3, respectively) are used in the experiment, as shown in Figure. S34(a). The total RF-to-RF efficiency (R#1+R#2+R#3) is also close to that in Figure. S32(b), showing that the transmitted power can be still properly split when more receiving antennas are considered. Both cases show that the proposed router has a good multiple-beam performance, that is, sufficiently delivering wireless power to multiple targets while keeping the total RF-to-RF efficiency steady.

Figure. S33. (a) The double-target experimental setup. (b) The measured double-target RF-to-RF efficiency.

Figure. S34. (a) The triple-target experimental setup. (b) The measured triple-target RF-to-RF efficiency.

(2) DC-to-DC Efficiency

The measured DC-to-DC efficiencies at different distances and with different DC loads are shown in Figure. S35. As can be seen, as the input DC power P_{DCin} increases, the DC-to-DC efficiencies first rise and then fall. This phenomenon results from the saturation effect of the diode used in the rectifier, as shown in Figure. S30. With the increase of the input RF power, the output DC voltage of the rectifier also rises until it reaches the minimum breakdown voltage (MBV) of the diode. When the output DC voltage surpasses the MBV, the diode operates in the saturation region and, in turn, lowers the entire WPT efficiency.

Figure. S35. The measured DC-to-DC efficiency. d is the distance between the router and the receiving antenna. R_{DC} is the DC load of the rectifier. (a) $d = 200$ mm; (b) $d = 400$ mm; (c) $d = 1000$ mm.

(3) Discussion on the WPT efficiency

The proposed WPT system has the best RF-to-RF of 3% and the best DC-to-DC efficiency (the entire WPT system) of 0.65%. Although the power transfer efficiency is on the order of several percent, it can still be comparable to recent state-of-the-art works [S3][S9]. Several reasons result in such an efficiency:

1) The diffraction-limited spot width. As the distance increases, the focal spot width increases according to the Abbe diffraction limitation by approximately

$$w = R \frac{\lambda}{D} \quad (6)$$

where R is the distance between the proposed router and the receiving antenna, λ is the wavelength, and D is the aperture size of the proposed router. On the other hand, the fixed receiving antenna's size results in the receiving antenna only capturing a partial portion of the delivered power. Diffraction-free beams that can propagate without diffractive spreading, such as Bessel beams, may improve the RF-to-RF efficiency. Another way to improve the efficiency can be increasing the aperture sizes of the router and the receiving antenna. The former can lead to smaller spot width according to Equation (6), while the latter enables capturing more delivered power.

2) The lossy substrates and diodes. The transmissive amplitudes of the RTM unit-cell are lower than unity due to the lossy substrates and diodes, as shown in Figure. S11. The advance of material science that enables substrates and diodes with lower losses can alleviate this problem.

3) The lossy cables and connectors. These cables and connectors are used to connect the router with the signal generator and the receiving antenna with the rectifier. Although lossless cables and connectors may not be available in practice, shortening or eliminating them is possible by integrating these components (including cables, connectors, the signal generator, et al.) into a chip so as to reduce these losses."

Comment: 10-Please comment on the switching speed. How dynamic a target could be for the system to be able to recognize the location and reconfigure the surface to focus the beam.

Response: We thank the reviewer for this comment. The switch speed is determined by three parts: the frame rate of the stereo camera, the detection speed of the model YOLOv5, and the beam switching speed.

The frame rate of the stereo camera and the detection speed of the model YOLOv5 are around 33 ms per frame (30 FPS) and 10 ms per frame (100 FPS), respectively. The beam switching speed can be estimated by the rising edge of the applied voltage to the unit-cell. As shown in Figure. R2 - 4, the duration of the rising edge is 0.88 us, and then the corresponding switching speed is $1 / (0.88 \text{ us}) = 1.136 \text{ MHz}$. The total switching time for each detection is $33 \text{ ms} + 10 \text{ ms} + 0.88 \text{ us} \approx 43 \text{ ms}$. For illustration, we take a target at 1m as an example. The 3dB beamwidth at 1m is $R\lambda/D = 1 * 0.052 / 0.3 = 0.173 \text{ m}$, where $R = 1 \text{ m}$ is the target distance, $\lambda = 0.052 \text{ m}$ is the operating wavelength, and $D = 0.3 \text{ m}$ is the size of the router. If we consider that the router is able to capture a target that moves a distance smaller than the 3dB beamwidth (0.173 m) within the total switching time (34 ms), the maximum target's speed can be calculated as $v = 0.173 / 0.043 = 4.02 \text{ m/s}$. A typical indoor target's speed is usually 1-2 m/s. Therefore, the proposed router has a promising switching speed for many practical scenarios.

From the above discussion, we can infer that the switching speed is mainly limited by the frame rate of the stereo camera and the detection speed of the model YOLOv5. The use of stereo cameras with higher frame rates and faster detection models can help improve the switching speed.

Figure. R2 - 4. The measured rising edge of the applied voltage

Comment: 11-Please explain the details of the beamforming algorithm.

Response: We thank the reviewer for this comment. Metasurfaces control incident electromagnetic (EM) waves by configuring the amplitude, phase, and polarization responses of all unit-cell. In our manuscript, the proposed metasurface controls EM waves in a phase-only manner, that is, by configuring the phase responses of the unit-cells. Thus, forming a beam or beams in space can be realized by synthesizing the phase responses of the unit-cell on the metasurfaces. The synthesis of phase responses (that is, the beamforming algorithm) is discussed in Supplementary Note 5 in the original manuscript. We have separated the beam algorithm as an independent part and supplemented more details on it. Please see the revised Supplementary Note 10.

The beamforming algorithm used in our work is the backward propagation algorithm, of which the effectiveness has been shown in other works [R1][R2]. Before explaining the backward propagation algorithm, we first introduce the forward propagation model of EM waves. In the forward propagation model, the metasurface can be regarded as an assembly of electrical dipoles. As shown in Figure. R2 - 5(a), the wavefront radiating from the metasurface can be written as

$$E(x, y, z) = \sum_n a_n \frac{e^{-jk\sqrt{(x-y_n')^2+(y-y_n')^2+(z-z_n')^2}}}{\sqrt{(x-x_n')^2+(y-y_n')^2+(z-z_n')^2}} \quad (1)$$

where a_n is the amplitude of the n th electrical dipole, k is the wave number, and the prime is used to indicate the source plane.

The backward propagation model propagates the wavefront backward to the source plane (in our case, the metasurface), as shown in Figure. R2 - 5(b). It is realized by numerically taking the conjugate of the exponential terms in Equation (1), that is,

$$E(x', y', z') = \sum_i w_i \frac{e^{jk\sqrt{(x_i-x')^2+(y_i-y')^2+(z_i-z')^2}}}{\sqrt{(x_i-x')^2+(y_i-y')^2+(z_i-z')^2}} \quad (2)$$

where i is the i th focus point and w_i is the intensity of the i th focus point.

Figure. R2 - 5. (a) The forward model. (b) The backward model.

The beamforming algorithm used in our work is explained as follows:

- (1) Input the laboratory coordinates of the detected objects $\{x_i, y_i, z_i\}$ and their power requirements $\{w_i\}$.
- (2) Calculate the interference pattern on the transmitter:

$$E = \sum_i w_i \frac{\exp\left(jk\sqrt{x_i^2 + y_i^2 + z_i^2}\right)}{\sqrt{x_i^2 + y_i^2 + z_i^2}}$$

and then extract the phase pattern:

$$\varphi = \arg(E) = \arg\left\{\sum_i w_i \frac{\exp\left(jk\sqrt{x_i^2 + y_i^2 + z_i^2}\right)}{\sqrt{x_i^2 + y_i^2 + z_i^2}}\right\}$$

- (3) Apply the phase pattern φ to the metasurface.

[R1] Brown T, Narendra C, Vahabzadeh Y, et al. On the use of electromagnetic inversion for metasurface design[J]. IEEE Transactions on Antennas and Propagation, 2019, 68(3): 1812-1824.

[R2] Chen K, Feng Y, Monticone F, et al. A reconfigurable active Huygens' metalens[J]. Advanced materials, 2017, 29(17): 1606422.

Revised Text:

In the main text:

"The synthesis of phase responses (or the beamforming algorithm) is detailed in Supplementary Note 10."

In the Supplementary Note 10:

"Reconfigurable metasurfaces control incident electromagnetic (EM) waves by configuring the unit-cell phase responses. Therefore, properly synthesizing the phase responses of reconfigurable metasurfaces enables the formation of desired single/multiple beams in space. The phase calculator, that is, the beamforming algorithm, is based on the backward propagation algorithm.

Once the laboratory coordinates of objects are predicted, the phase patterns of the transmitter can be calculated according to the backward propagation algorithm, which regards objects as point sources and extracts the phase of the interference pattern on the transmitter. The phase calculator, that is, the beamforming algorithm, used in our work is described as follows:

- 1) Input the laboratory coordinates of the detected objects $\{x_i, y_i, z_i\}$ and their power requirements $\{w_i\}$.
- 2) Calculate the interference pattern on the transmitter:

$$E = \sum_i w_i \frac{\exp\left(jk\sqrt{x_i^2 + y_i^2 + z_i^2}\right)}{\sqrt{x_i^2 + y_i^2 + z_i^2}} \quad (5)$$

and then extract the phase pattern:

$$\varphi = \arg(E) = \arg\left\{\sum_i w_i \frac{\exp\left(jk\sqrt{x_i^2 + y_i^2 + z_i^2}\right)}{\sqrt{x_i^2 + y_i^2 + z_i^2}}\right\} \quad (6)$$

- 3) Apply the phase pattern φ to the metasurface."

Comment: 12-Please check the manuscript for formatting errors. Please correct “unman” to “unmanned”.

Response: We thank the reviewer for this kind reminder. "unman" has been revised into "unmanned" in revised text and figures. We have checked our manuscript carefully and corrected the typos.

Revised Text:

In the main text:

"environment" → "environments"

"unman devices" → "unmanned devices"

"reciever" → "receiver", and et al.

In Figure 1:

Responses to Reviewer #3:

Comment: Reviewer #3 (Remarks to the Author):

The manuscript presents "Intelligent Wireless Power Transfer via a 2-Bit Compact Reconfigurable Transmissive-Metasurface-Based Router " which is a well-organized and technically sound analysis of the topic at hand. However, the findings in this manuscript do not appear to introduce any significant novelty to the field. To enhance the impact of this work, I suggest the authors consider the following comments.

Response: We are grateful for the reviewer's comments. We would like to explain the novelty of this work.

Firstly, we propose a compact, reconfigurable transmissive-metasurface-based framework. The core components of the proposed framework include a plane-wave feeder and a transmissive 2-bit reconfigurable metasurface-based beam generator, which constitute a reconfigurable power router with a total thickness of 0.8λ . The profile compression renders the proposed framework more sufficient for applications with miniaturization and integration requirements, such as indoor environments. The low cost and compact size of the proposed framework may boost the commercialization of metasurface-based systems.

Secondly, we experimentally show that the propose framework is capable of on-demand wireless power delivery in random environments. The advancements of the Internet of Things (IoT) bring new challenges for wireless power transfer (WPT). IoT devices like unmanned delivery car and small sensors coexist in sophisticated electromagnetic environments. The indeterminacy of the quantity, location, and power consumption is also an intractable problem. Therefore, it is crucial to present flexible strategies to selectively charge multiple devices in environments where power-consuming devices appear/disappear randomly. In this manuscript, we provide a flexible strategy for multiple-device WPT that enables object detection, localization and tracking, and power delivery to multiple autonomous devices. We experimentally demonstrated that the proposed framework is capable of power delivery to selected

moving or static devices in unpredictable environments. The experimental results show that the proposed framework can be a promising solution for multiple-device WPT in dynamic environments.

Thirdly, we provide a potential solution for the fusion of WPT systems and wireless information transfer (WIT) systems. Large-scale device interconnections in IoT create new demands for the design of WPT and WIT systems. Metasurface-based WPT and WIT are two emerging strategies for IoT and sixth-generation (6G) communication, one for power only and the other for communication only. In our manuscript, we experimentally show that the proposed framework enables simultaneous wireless information and power transfer, providing a potential solution for the smoothly integrated design of WPT and WIT systems. The fusion of WPT and WIT systems is a significant strategy to take full advantage of the electromagnetic spectrum.

In summary, the proposed framework is beneficial for IoT and 6G applications in sophisticated electromagnetic environments. The compactness, reconfigurability, and intelligence of the proposed framework hold immense potential for the commercialization of cost-effective and complexity-reduced metasurface-based systems.

Comment: 1. In the introduction, there is not enough information about the studies on self-adaptive metasurfaces and why this research is needed. The research on programmable and self-adaptive metasurfaces have been actively studied. It would be better to provide them as references.

Response: We thank the reviewer for this important comment. We have provided more discussion on why this research is needed in the revised manuscript. The references to programmable and self-adaptive metasurfaces are also supplemented in the revised manuscript.

Revised Text:

i) In the main text:

"Another area of significant interest is programmable metasurfaces [36], which utilize

digitally programmable unit-cells to provide a link between the physical and digital worlds [37][38]. Thanks to their prominent characteristics, programmable metasurfaces have found applications in various domains, including polarization controls [39], multi-frequency modulations [40][41][42], and meta-imagers [44][45]. Additionally, optically driven metasurfaces are proposed, allowing for microwave manipulation using light [46][47][48]. Self-adaptive metasurfaces have also been extensively studied [49][50], showcasing their potential in many scenarios, such as beam steering [51], cloaks [52], and dynamic reactions [53]. "

ii) In the main text:

"Despite these successfully demonstrated metasurfaces, complex environments like indoors and industries cultivate new needs for metasurface-based WPT systems. Firstly, the coexistence of large amounts of IoT devices, such as unmanned delivery cars and small sensors, in sophisticated electromagnetic environments becomes an intractable problem for the WPT systems. Flexible charging strategies for selected power-consuming devices are crucial in environments where multiple devices appear and disappear at random. In addition, in practice, finite spaces necessitate device miniaturization and integration. Conventional feeders used to excite metasurfaces, such as horn antennas, suffer from nonuniform wave fronts, resulting in large system profiles on the order of several to ten wavelengths. The large profiles make them challenging for applications with requirements for miniaturization and integration. Furthermore, with the large deployments of IoT devices, wireless spectrum bands, like unlicensed bands, have seen increasingly crowded usage. Since electromagnetic waves can simultaneously carry energy and information, there is a need for the fusion of WPT systems and wireless information transfer (WIT) systems to take full advantage of the wireless spectrums. This fusion may also help to address the spectrum shortage problem."

Comment: 2. Based on my limited knowledge, as for the terminology "SWPIT", it is commonly used as simultaneous wireless information and power transfer rather than

simultaneous wireless power and information transfer. I recommend using simultaneous wireless power and information transfer (SWIPT) for consistency.

Response: We thank the reviewer for this important comment. The "SWIPT" is commonly used. We have modified "SWPIT" as "SWIPT" in the revised manuscript.

Revised Text:

In the main text:

"we also demonstrate that the proposed framework enables simultaneous wireless information and power transfer (SWIPT)"

"SWPIT" → "SWIPT"

In the Supplementary Note 13:

"Supplementary Note 13: Simultaneous Wireless Information and Power Transfer (SWIPT)"

Comment: 3. It would be good if you can provide the design procedure of the metasurface unit cell. The reason for choosing that specific design, etc.

Response: We thank the reviewer for this important comment. The design procedure and evolution of the metasurface unit-cell has been provided in the revised Supplementary Note 3.

Revise Text:

In the Supplementary Note 3:

"Here we describe the design procedure and evolution of the 2-bit reconfigurable-transmissive-metasurface (RTM) unit-cell.

Firstly, we would like to determine the basic geometry as the initial model for the RTM unit-cell. According to the transmit-array theory [S7], a single-layer structure that can only support tangential electric currents radiates equally on both sides. To manipulate transmissive waves with minimized reflections and transmissive phases over the 2π range, multi-layer structures should be considered. Adopted from the transmit-array, the transmitter-receiver architecture is used in our initial model for the RTM unit-cell. Due to the wideband and simple geometry, a U-slot antenna is selected

as the transmitting and receiving antennas, as shown in Figure. S7(a). We perform a symmetry operation on the U-slot antenna and acquire an O-slot antenna with two diodes loaded, as shown in Figure. S7(b). The 180° phase manipulation is obtained by switching the states of the two diodes: when Diode#1 is ON and Diode#2 is OFF, the current flows from upward to downward, as shown in Figure. S7(c); when Diode#1 is OFF and Diode#2 is ON, the current flows from downward to upward, as shown in Figure. S7(d). We can obtain the transmitter-receiver architecture by performing a further symmetry operation, as shown in Figure. S7(e). As can be seen, the transmitter-receiver architecture is naturally a multi-layer structure, which can support high transmissions and a large transmissive phase range. A central via is used to connect the transmitter with the receiver, enabling the power coupling from the receiver to the transmitter.

Secondly, a 90° delay line is used to acquire another two phase levels (that is, 90° and 270°), as shown in Figure. S8(a). As we have explained in the main text, the additional 90° delay line is needed to realize the 2-bit phase manipulation. When Diode#3 is ON and Diode#4 is OFF, the current will flow through the delay line, obtaining an additional 90° phase, as shown in Figure. S8(b).

Thirdly, DC lines are added to provide control signals to the diode states. Besides, several structures that prevents incident high-frequency signals from entering the control circuit board are also needed, as shown in Figure. S10 (d) and (f). On the Rx side, two crescent metal patches are used to constitute large capacitors with the Rx ground plane because of the simple geometry (less geometry parameters) and high symmetry, which is beneficial for the parameter optimization. Due to the delay line, the transmitter loses its symmetry. Thus, two fan branches are used to constitute large capacitors with the Tx ground plane. When entering the DC lines, incident high-frequency signals see ground due to the large capacitors constituted by the crescent metal patches and the fan branches, and thus are blocked from the control circuit board.

Figure. S7. The basic geometry of the RTM unit-cell.

Figure. S8. The delay line is added. (a) 3D view. (b) Lateral view."

Comment: 4. Design, simulation, and analysis of the rectifier part are missing.

Response: We thank the reviewer for this important comment. We have supplemented the design, simulation, and analysis of the rectifier part in the revised Supplementary Note 7.

Revised Text:

In the Supplementary Note:

"Supplementary Note 7: Rectifier

A rectifier is a rectifying circuit, which converts received microwave powers into DC

power. As shown in Figure. S27, a rectifier usually consists of three components: a matching circuit, a diode, and a filter circuit. The matching circuit matches the input impedance with the subsequent circuits. The diode is used to convert AC to DC by allowing current to pass in only one direction. The filter circuit filters out the fundamental wave and all harmonics, leaving only direct current at the output port.

Figure. S27. The components of a rectifier.

The designed rectifier is shown in Figure. S28. The matching circuit adopts a single short-circuit stub and provides a DC-to-ground loop. By adjusting the length of the short-circuit sub, the input impedance can match with the subsequent circuits. The rectifier works at 5.8GHz. Therefore, it is necessary to filter out the harmonic components of 5.8 GHz, 11.6 GHz, and 17.4 GHz because the magnitudes of other higher harmonics are small. A band-stop filter with two open-circuit stubs is used as the filter circuit, with lengths of $\lambda/4$ and $\lambda/8$ at 5.8 GHz, respectively. The simulated results of the filter circuit are shown in Figure. S29. As can be seen, the transmission S21 at 5.8 GHz, 11.6 GHz, and 17.4 GHz is lower than -24 dB, indicating that the harmonics are effectively filtered out from the output port.

Figure. S28. The designed rectifier.

Figure. S29. The simulated results of the filter circuit.

Figure. S30 (a) shows the detailed dimensions of the rectifier. The relative permittivity of the substrate is 4.0 with $\tan\delta=0.001$. The Schottky diode is BAT1705 from Infineon Technologies. The optimized resistor is $400\ \Omega$. The measured results of the rectifier are shown in Figure. S30 (b) and (c). The rectifier efficiency is defined as $\eta = P_{in}/P_{DC}$, where P_{in} is the input RF power and P_{DC} is the output DC power.

Figure. S30. (a) The detailed dimensions of the rectifier. The measured results of the rectifier: (b) The rectifier efficiency at 5.8GHz; (c) The maximum rectifier efficiency over frequency.

Comment: 5. Lack of analysis on transfer efficiencies.

Analysis on the Tx to Rx (single and multi-receivers), Rx-load (RF-DC conversion), and entire WPT efficiencies are required.

Response: We thank the reviewer for this comment. We have supplemented some analysis on the power transfer efficiencies. The RF-DC conversion efficiencies have been supplemented in the revised Supplementary Note 7. The Tx to Rx and entire WPT efficiencies have been also supplemented. Please see the revised Supplementary Note 8.

Revised Text:

In the Supplementary Note 8

" Supplementary Note 8: WPT efficiencies

The entire WPT system consists of a commercial tunable signal generator, the proposed router, a receiving antenna, and a rectifier, as illustrated in Figure. S31. To fully characterize the entire system, the DC-to-DC efficiency of all the components, from the DC input to the DC output, can be considered as follows:

$$\eta_{DD} = \eta_{DR} \eta_{RR} \eta_{RD} \quad (4)$$

where η_{DR} is the efficiency of the commercial tunable signal generator (or DC-to-RF efficiency), η_{RR} is the RF-to-RF efficiency, and η_{RD} is the rectifier efficiency (or RF-to-DC efficiency). In our proposed system, η_{DR} has a typical value of 40%. The rectifier efficiency η_{RD} has been shown in Figure. S30. In the following, we focus on the RF-to-RF efficiency and the DC-to-DC efficiency.

Figure. S31. The efficiency of the WPT system.

(1) RF-to-RF Efficiency

The experimental setup is shown in Figure. S31. The antennas shown in Supplementary Note 6 are used as the receiving antennas in this section. We first measured the RF-to-RF efficiency between the NFF and the receiving antenna (denote as the NFF group). Next, we inserted the DBG to constitute the router and measured the corresponding RF-to-RF efficiency (denote as the NFF+DBG group). To illustrate the improvement after the insertion of the DBG, we define the beam gain as

$$G = 10 \log_{10} \left(\frac{P_{ND}}{P_N} \right) \quad (5)$$

where P_{ND} is the received power of the NFF+DBG group and P_N is the received power of the NFF group. The corresponding results are shown in Figure. S32.

(1.1) Single-Target RF-to-RF Efficiency

Figure. S32 shows the single-target RF-to-RF efficiency. In this case, only a single receiving antenna is used. As can be seen from Figure. S32(a) and (b), as the distance d increases, the RF-to-RF efficiency decreases. This is due to the spreading nature of electromagnetic waves. From Figure. S32(a), for $\theta = 0$, the RF-to-RF efficiency is lowered after the insertion of the DBG. This results from the transmissive amplitude of the RTM being not unity, as shown in Figure 3(d) and (e) in the main text. As shown in Figure. S32(c), the average gain (loss) is -3.72 (3.72) dB, which is close to the measured transmissive amplitudes of the RTM shown in Figure 3(d) and (e) in the main text. In contrast, for $\theta = 45^\circ$, the RF-to-RF efficiency is improved after the insertion of the DBG, as shown in Figure. S32(b). The average gain is 13.41 dB. The insertion of the DBG enables the reconfigurable beams to redirect the transmitted powers to the receiving antenna and, in turn, improves the RF-to-RF efficiency while the receiver is placed at off-axis positions.

Figure. S32. The measured single-target RF-to-RF efficiency: (a) $\theta = 0^\circ$; (b) $\theta = 45^\circ$. (c) Gain after

inserting the DBG. NFF represents the nearfield feeder, and DBG represents the dynamic beam generator.

(1.2) Multiple-Target RF-to-RF Efficiency

We also measured the multiple-target RF-to-RF efficiency. In the first case, the experimental setup with two receiving antennas (denoted as R#1 and R#2, respectively) is considered. The two receiving antennas are placed at -45° and $+45^\circ$, respectively, as shown in Figure. S33(a). The corresponding measured RF-to-RF efficiencies are shown in Figure. S33(b). We can see that the total RF-to-RF efficiency (R#1+R#2) in Figure. S33(b) is close to that of single target with $\theta = 45^\circ$ in Figure. S32(b). This indicates that the transmitted power is properly split into two parts, one for R#1 and the other for R#2. In the second case, three receiving antennas (denoted as R#1, R#2, and R#3, respectively) are used in the experiment, as shown in Figure. S34(a). The total RF-to-RF efficiency (R#1+R#2+R#3) is also close to that in Figure. S32(b), showing that the transmitted power can be still properly split when more receiving antennas are considered. Both cases show that the proposed router has a good multiple-beam performance, that is, sufficiently delivering wireless power to multiple targets while keeping the total RF-to-RF efficiency steady.

Figure. S33. (a) The double-target experimental setup. (b) The measured double-target RF-to-RF efficiency.

Figure. S34. (a) The triple-target experimental setup. (b) The measured triple-target RF-to-RF efficiency.

(2) DC-to-DC Efficiency

The measured DC-to-DC efficiencies at different distances and with different DC loads are shown in Figure. S35. As can be seen, as the input DC power P_{DCin} increases, the DC-to-DC efficiencies first rise and then fall. This phenomenon results from the saturation effect of the diode used in the rectifier, as shown in Figure. S30. With the increase of the input RF power, the output DC voltage of the rectifier also rises until it reaches the minimum breakdown voltage (MBV) of the diode. When the output DC voltage surpasses the MBV, the diode operates in the saturation region and, in turn, lowers the entire WPT efficiency.

Figure. S35. The measured DC-to-DC efficiency. d is the distance between the router and the receiving antenna. R_{DC} is the DC load of the rectifier. (a) $d = 200$ mm; (b) $d = 400$ mm; (c) $d = 1000$ mm.

(3) Discussion on the WPT efficiency

The proposed WPT system has the best RF-to-RF of 3% and the best DC-to-DC efficiency (the entire WPT system) of 0.65%. Although the power transfer efficiency is on the order of several percent, it can still comparable to recent state-of-the-art works

[S3][S9]. Several reasons result in such an efficiency:

1) The diffraction-limited spot width. As the distance increases, the focal spot width increases according to the Abbe diffraction limitation by approximately

$$w = R \frac{\lambda}{D} \quad (6)$$

where R is the distance between the proposed router and the receiving antenna, λ is the wavelength, and D is the aperture size of the proposed router. On the other hand, the fixed receiving antenna's size results in the receiving antenna only capturing a partial portion of the delivered power. Diffraction-free beams that can propagate without diffractive spreading, such as Bessel beams, may improve the RF-to-RF efficiency. Another way to improve the efficiency can be increasing the aperture sizes of the router and the receiving antenna. The former can lead to smaller spot width according to Equation (6), while the latter enables capturing more delivered power.

2) The lossy substrates and diodes. The transmissive amplitudes of the RTM unit-cell are lower than unity due to the lossy substrates and diodes, as shown in Figure. S11. The advance of material science that enables substrates and diodes with lower losses can alleviate this problem.

3) The lossy cables and connectors. These cables and connectors are used to connect the router with the signal generator and the receiving antenna with the rectifier. Although lossless cables and connectors may not be available in practice, shortening or eliminating them is possible by integrating these components (including cables, connectors, the signal generator, et al.) into a chip so as to reduce these losses."

Comment: 6. How and why the separation between NFF-DBG is selected to be 13 mm? Normally, a certain separation is required for the metasurface superstrate. It would be great if the authors can provide some scientific evidence on how could the authors reduce the distance between them as it is related to the main contribution (compact size) of this manuscript.

Response: We thank the reviewer for this comment. The separation between the NFF

and the DBG is carefully selected to ensure efficient power injection into the router while keeping the resonant frequency at 5.8GHz. Figure R3 - 1 shows the reflection coefficient (S_{11}) of the router under different NFF-DBG separations. As can be seen, as the separation is greater or equal to 13 mm, the resonant frequency remains stable at 5.8GHz. The DBG exhibits high transmittance with a relatively low equivalent impedance. The coupling between the NFF and the DBG is relatively weak, resulting in small impact on the system's performance. However, as the separation is further reduced, the coupling intensifies, leading to a more significant influence on the S_{11} . This, in turn, results in a notable shift in the system's resonant characteristics, ultimately compromising system performance. Consequently, we have selected a NFF-DBG separation of 13 mm to ensure the router consistently resonates at 5.8GHz.

Figure R3 - 1. The resonant characteristics of the router under different NFF-DBG separations. 'RFF' refers to the resonant characteristics of the NFF.

Responses to Reviewer #4:

Comment: Reviewer #4 (Remarks to the Author):

The paper proposes a compact reconfigurable power router that consists of a plane-wave feeder and a transmissive 2-bit reconfigurable metasurface-based beam generator. The paper is well organized and written with several experiment results. However, there are still concerns that need to be addressed as follows:

Response: We appreciate the reviewer's valuable comments. The insightful comments are very helpful in further refining our manuscript. In the following, we address the specific comments point-by-point while revising our manuscript.

Comment: 1. Please clearly emphasize the novelty of this work compared to the existing works.

Response: We are grateful for the reviewer's comments. We would like to emphasize the novelty of this work.

We are grateful for the reviewer's comments. We would like to explain the novelty of this work.

Firstly, we propose a compact, reconfigurable transmissive-metasurface-based framework. The core components of the proposed framework include a plane-wave feeder and a transmissive 2-bit reconfigurable metasurface-based beam generator, which constitute a reconfigurable power router with a total thickness of 0.8λ . The profile compression renders the proposed framework more sufficient for applications with miniaturization and integration requirements, such as indoor environments. The low cost and compact size of the proposed framework may boost the commercialization of metasurface-based systems.

Secondly, we experimentally show that the propose framework is capable of on-demand wireless power delivery in random environments. The advancements of the Internet of Things (IoT) bring new challenges for wireless power transfer (WPT). IoT

devices like unmanned delivery car and small sensors coexist in sophisticated electromagnetic environments. The indeterminacy of the quantity, location, and power consumption is also an intractable problem. Therefore, it is crucial to present flexible strategies to selectively charge multiple devices in environments where power-consuming devices appear/disappear randomly. In this manuscript, we provide a flexible strategy for multiple-device WPT that enables object detection, localization and tracking, and power delivery to multiple autonomous devices. We experimentally demonstrated that the proposed framework is capable of power delivery to selected moving or static devices in unpredictable environments. The experimental results show that the proposed framework can be a promising solution for multiple-device WPT in dynamic environments.

Thirdly, we provide a potential solution for the fusion of WPT systems and wireless information transfer (WIT) systems. Large-scale device interconnections in IoT create new demands for the design of WPT and WIT systems. Metasurface-based WPT and WIT are two emerging strategies for IoT and sixth-generation (6G) communication, one for power only and the other for communication only. In our manuscript, we experimentally show that the proposed framework enables simultaneous wireless information and power transfer, providing a potential solution for the smoothly integrated design of WPT and WIT systems. The fusion of WPT and WIT systems is a significant strategy to take full advantage of the electromagnetic spectrum.

In summary, the proposed framework is beneficial for IoT and 6G applications in sophisticated electromagnetic environments. The compactness, reconfigurability, and intelligence of the proposed framework hold immense potential for the commercialization of cost-effective and complexity-reduced metasurface-based systems.

Revised Text:

In the main text:

" Despite these successfully demonstrated metasurfaces, complex environments like indoors and industries cultivate new needs for metasurface-based WPT systems. Firstly, the coexistence of large amounts of IoT devices, such as unmanned delivery cars and

small sensors, in sophisticated electromagnetic environments becomes an intractable problem for the WPT systems. Flexible charging strategies for selected power-consuming devices are crucial in environments where multiple devices appear and disappear at random. In addition, in practice, finite spaces necessitate device miniaturization and integration. Conventional feeders used to excite metasurfaces, such as horn antennas, suffer from nonuniform wave fronts, resulting in large system profiles on the order of several to ten wavelengths. The large profiles make them challenging for applications with requirements for miniaturization and integration. Furthermore, with the large deployments of IoT devices, wireless spectrum bands, like unlicensed bands, have seen increasingly crowded usage. Since electromagnetic waves can simultaneously carry energy and information, there is a need for the fusion of WPT systems and wireless information transfer (WIT) systems to take full advantage of the wireless spectrums. This fusion may also help to address the spectrum shortage problem.

In this article, we propose a subwavelength WPT framework, which consists of a planar-wave feeder, a transmissive 2-bit reconfigurable metasurface-based beam generator, an environment sensor, and a computation unit. The plane-wave feeder and the beam generator create a compact reconfigurable power router with a total thickness of 0.8 wavelength. The subwavelength profile is an order of magnitude smaller than that with conventional feeders, making it suitable for system miniaturization and integration. Cooperating with the sensor capturing environmental information, the computation unit is able to control the reconfigurable power router in real-time to deliver wireless power to single or multiple targets in stochastic environments. Experiments demonstrate that the framework has the capability of detecting and localizing multiple targets, and then selectively supplying power to those targets according to their energy requirements. In addition, we also demonstrate that the proposed framework enables simultaneous wireless information and power transfer (SWIPT), providing a possible remedy for the fusion of WPT systems and WIT systems. The proposed framework holds immense application potential in dynamic environments with wireless devices like sensors, smart home devices, and industrial delivery robots."

Comment: 2. The authors claimed that the proposed reconfigurable power router has a total thickness of 0.8 wavelength. However, as shown in the global view of the fabricated WPT system in Figure. S9, the overall size of the router with the control board is bigger. For a fair comparison with the existing works, it is recommended to correct the total thickness of the proposed framework since, without the control board, the router could not work.

Response: We thank the reviewer for this comment. The control board is an external component. It can be miniaturized by adopting customized integrated voltage chips, but it is not the focus of our work. Commonly, the profiles of current work on metasurface-based systems include the thickness of the metasurface, the profile of the feeder, and the separation between the metasurface and the feeder, while the size of the control board is excluded. Therefore, as we compare the profile of the proposed systems with other works, the thickness of the metasurface, the profile of the feeder, and the separation between the metasurface and the feeder are considered, but the control board is excluded. Please see these recently proposed metasurface-based systems, which only consider the separation between the metasurfaces and the feeders [R1][R2][R3].

[R1] Li W, Ma Q, Liu C, et al. Intelligent metasurface system for automatic tracking of moving targets and wireless communications based on computer vision[J]. Nature Communications, 2023, 14(1): 989.

[R2] Han J, Li L, Ma X, et al. Adaptively smart wireless power transfer using 2-bit programmable metasurface[J]. IEEE Transactions on Industrial Electronics, 2021, 69(8): 8524-8534.

[R3] Zheng Y, Chen K, Xu Z, et al. Metasurface-Assisted Wireless Communication with Physical Level Information Encryption[J]. Advanced Science, 2022, 9(34): 2204558.

Comment: 3. Please thoroughly elaborate on how to design the artificial magnetic conductor (AMC) and partial reflected plane (PRP). How can the authors calculate the

dimensions for AMC and PRP? If there is any reference, please cite it.

Response: We thank the reviewer for this comment. There is no sufficiently accurate method to analytically calculate the dimensions for AMC and PRP. Determining the geometries and the corresponding dimensions for AMC and PRP usually relies on full-wave simulations. Therefore, we would like to provide a detailed design procedure on how the dimensions for AMC and PRP are determined.

Firstly, let's consider the AMC design. AMC is a metallic array printed on a grounded dielectric substrate that can fully reflect incident waves with a near-zero reflection phase [R1][R2]. A typical unit-cell of AMC is a square patch printed on a grounded dielectric, as shown in Figure. R4 - 1(a). The analysis of the unit-cell adopts the equivalent circuit model, where the square patch is represented by an inductor and a capacitor, the substrate is represented by a uniform transmission line, and the ground is represented by a short circuit. Adjusting the dimensions of the rectangle patch and the substrate is equivalent to adjusting the inductance, the capacitance, and the length of the transmission line. Accordingly, the reflection coefficient $\Gamma = E_r/E_i$ is adjusted, where E_i is the incident wave and E_r is the reflection wave. Figure. R4 - 1(b) shows the simulated reflection of the AMC unit-cell. As can be seen, by properly adjusting the side length of the square patch, the reflective phase of the AMC unit-cell can be near zero degrees. The reflective amplitude remains near one, showing the full reflection property.

The design procedure for PRP is similar to that of AMC. Since partial reflection is considered rather than full reflection, we replace the ground in AMC with a defected ground (a square ring patch) to allow power leakage, as shown in Figure. R4 - 2(a). The reflection properties of PRP unit-cell is shown in Figure. R4 - 2(b). Since the PRP doesn't require full reflection, the reflective amplitude can be lower than 1. The reflective phase remains as a degree of freedom for the final optimization.

Finally, the full-wave simulation is carried out by CST Microwave Studio for the final optimization. The final optimized parameters are shown in Supplementary Note 2, and the corresponding simulated and measured results are shown in the main text.

Figure. R4 - 1. (a) The unit-cell of AMC. (b) The reflection properties. l is the side length of the square patch.

Figure. R4 - 2. (a) The unit-cell of PRP. (b) The reflection property.

[R1] Erdemli Y E, Sertel K, Gilbert R A, et al. Frequency-selective surfaces to enhance performance of broad-band reconfigurable arrays[J]. IEEE Transactions on Antennas and Propagation, 2002, 50(12): 1716-1724.

[R2] Feresidis A P, Goussetis G, Wang S, et al. Artificial magnetic conductor surfaces and their application to low-profile high-gain planar antennas[J]. IEEE Transactions on Antennas and Propagation, 2005, 53(1): 209-215.

Comment: 4. What is the transmitted power in the experiment? Please clarify it.

Response: We thank the reviewer for this comment. The transmitted power, i.e., output RF power of the tunable signal generator, ranges from 1 W to 50 W. The transmitted power in the experiments is altered in accordance with the target distance and power consumption.

Comment: 5. In the first experiment, the authors show that the proposed WPT can light up the LED, which only consumes several mW. However, in the second experiment, they attempted to charge daily-life electronic devices (i.e., a smartphone and a power bank). The power consumption of these devices is relatively high, about 2 to 6 W. Transferring such a high RF power is challenging in WPT system. Please clarify the transmitted power and the received power at these focal spots.

Response: We thank the reviewer for this comment. Indeed, the power consumption of the LED is on the order of several mW, while the power consumption of the used electronic devices is several W. The maximum output voltage of the fabricated rectifier is 5.09 V, which is a typical voltage value for many daily-life electronic devices and thus enables charging these devices. To determine the transmitted power for a certain device, we measured the RF-to-RF efficiency between the router and the receiving antenna is measured. These measurements were conducted at various distances and with different numbers of targets. The measured results are shown in Figure. R4 - 3. The total RF-to-RF efficiencies ($R\#1+R\#2$ and $R\#1+R\#2+R\#3$), as depicted in Figure. R4 - 3(b) and (c), are close to that of a single target with $\theta = 45^\circ$, as shown in Figure. R4 - 3(a). This indicates the router's good performance in handling multiple beams. Therefore, when two or three targets need to be charged simultaneously, the corresponding transmitted power increases by over two or threefold, respectively, to ensure the power requirements are met for multiple targets. This pattern continues for additional targets.

Figure. R4 - 3. The measured RF-to-RF efficiency. (a) Single target. (b) Double targets. (c) Triple targets. The targets are denoted as R#1, R#2, and R#3, respectively. The top row corresponds to the experimental setups. The bottom row corresponds to the measured results.

In the experiments, the distance between the proposed router and the power-consuming device ranges from 700 mm to 800 mm. According to Figure. R4 - 3, we can extract the corresponding RF-to-RF efficiency, which ranges from 0.2% to 0.4% with single beam considered. The transmitted power can be calculated as follows:

1) The ON voltage of the LED used in our experiments is 1.8 V. To light up such an LED, the input power to the rectifier is 16 mW (the corresponding output DC voltage of the rectifier is 2 V). Therefore, the transmitted power can be calculated as $0.016 / 0.002 = 8$ W. When two LEDs are considered, the transmitted power doubles.

2) To charge the smartphone and the power bank, the output DC voltage of the rectifier should be 5 V. The corresponding input RF power is 100 mW (20 dBm), as shown in Supplementary Note 5. Therefore, the transmitted power can be calculated as $0.1 / 0.002 = 50$ W.

We admit that the power consumption may be greater than the delivered power for these electronic devices (the smartphone and the power bank). This is restricted by the available DC output. The maximum output DC power of the fabricated rectifier is 65 mW (please see Supplementary Note 7), which, even so, is comparable to these state-of-the-art rectifiers [R1][R2]. The limited output DC power results from the performance of commercially-available Schottky diodes [R3]. The power level of the

fabricated rectifier is relatively low but its output voltage is large enough to charge the electronic devices demonstrated in our manuscript. Although there is research on the Watt-level rectifier circuit [R4], the diodes in these studies are customized and still not commercially available. A transitional method before the availability of higher-power rectifiers is introducing power management modules, which enable power storage in 'idle' time.

[R1] Zhang S, Zhu J, Zhang Y, et al. Standalone stretchable RF systems based on asymmetric 3D microstrip antennas with on-body wireless communication and energy harvesting[J]. *Nano Energy*, 2022, 96: 107069.

[R2] Lu P, Song C, Huang K M. A compact rectenna design with wide input power range for wireless power transfer[J]. *IEEE Transactions on Power Electronics*, 2020, 35(7): 6705-6710.

[R3] Ngo T, Huang A D, Guo Y X. Analysis and design of a reconfigurable rectifier circuit for wireless power transfer[J]. *IEEE Transactions on Industrial Electronics*, 2018, 66(9): 7089-7098.

[R4] Dang K, Zhang J, Zhou H, et al. A 5.8-GHz high-power and high-efficiency rectifier circuit with lateral GaN Schottky diode for wireless power transfer[J]. *IEEE Transactions on Power Electronics*, 2019, 35(3): 2247-2252.

Comment: 6. What is the power transfer efficiency (PTE) of the system? It would be better to include a figure of PTE over the target moving distance to show the effectiveness of the proposed router.

Response: We thank the reviewer for this comment. Following the advice of the reviewer, we have supplemented the measured PTE in our revised manuscript for more complete discussion. The PTE is measured under different distances and different angles. Please see the revised Supplementary Note 8.

Revised Text:

In the Supplementary Note 8

" Supplementary Note 8: WPT efficiencies

The entire WPT system consists of a commercial tunable signal generator, the proposed router, a receiving antenna, and a rectifier, as illustrated in Figure. S31. To fully characterize the entire system, the DC-to-DC efficiency of all the components, from the DC input to the DC output, can be considered as follows:

$$\eta_{DD} = \eta_{DR} \eta_{RR} \eta_{RD} \quad (4)$$

where η_{DR} is the efficiency of the commercial tunable signal generator (or DC-to-RF efficiency), η_{RR} is the RF-to-RF efficiency, and η_{RD} is the rectifier efficiency (or RF-to-DC efficiency). In our proposed system, η_{DR} has a typical value of 40%. The rectifier efficiency η_{RD} has been shown in Figure. S30. In the following, we focus on the RF-to-RF efficiency and the DC-to-DC efficiency.

Figure. S31. The efficiency of the WPT system.

(1) RF-to-RF Efficiency

The experimental setup is shown in Figure. S31. The antennas shown in Supplementary Note 6 are used as the receiving antennas in this section. We first measured the RF-to-RF efficiency between the NFF and the receiving antenna (denote as the NFF group). Next, we inserted the DBG to constitute the router and measured the corresponding RF-to-RF efficiency (denote as the NFF+DBG group). To illustrate the improvement after the insertion of the DBG, we define the beam gain as

$$G = 10 \log_{10} \left(\frac{P_{ND}}{P_N} \right) \quad (5)$$

where P_{ND} is the received power of the NFF+DBG group and P_N is the received power of the NFF group. The corresponding results are shown in Figure. S32.

(1.1) Single-Target RF-to-RF Efficiency

Figure. S32 shows the single-target RF-to-RF efficiency. In this case, only a single receiving antenna is used. As can be seen from Figure. S32(a) and (b), as the distance d increases, the RF-to-RF efficiency decreases. This is due to the spreading nature of electromagnetic waves. From Figure. S32(a), for $\theta = 0$, the RF-to-RF efficiency is lowered after the insertion of the DBG. This results from the transmissive amplitude of the RTM being not unity, as shown in Figure 3(d) and (e) in the main text. As shown in Figure. S32(c), the average gain (loss) is -3.72 (3.72) dB, which is close to the measured transmissive amplitudes of the RTM shown in Figure 3(d) and (e) in the main text. In contrast, for $\theta = 45^\circ$, the RF-to-RF efficiency is improved after the insertion of the DBG, as shown in Figure. S32(b). The average gain is 13.41 dB. The insertion of the DBG enables the reconfigurable beams to redirect the transmitted powers to the receiving antenna and, in turn, improves the RF-to-RF efficiency while the receiver is placed at off-axis positions.

Figure. S32. The measured single-target RF-to-RF efficiency: (a) $\theta = 0^\circ$; (b) $\theta = 45^\circ$. (c) Gain after inserting the DBG. NFF represents the nearfield feeder, and DBG represents the dynamic beam generator.

(1.2) Multiple-Target RF-to-RF Efficiency

We also measured the multiple-target RF-to-RF efficiency. In the first case, the experimental setup with two receiving antennas (denoted as R#1 and R#2, respectively) is considered. The two receiving antennas are placed at -45° and $+45^\circ$, respectively, as shown in Figure. S33(a). The corresponding measured RF-to-RF efficiencies are shown in Figure. S33(b). We can see that the total RF-to-RF efficiency (R#1+R#2) in Figure. S33(b) is close to that of single target with $\theta = 45^\circ$ in Figure. S32(b). This indicates that the transmitted power is properly split into two parts, one for R#1 and the other for R#2.

In the second case, three receiving antennas (denoted as R#1, R#2, and R#3, respectively) are used in the experiment, as shown in Figure. S34(a). The total RF-to-RF efficiency ($R\#1+R\#2+R\#3$) is also close to that in Figure. S32(b), showing that the transmitted power can be still properly split when more receiving antennas are considered. Both cases show that the proposed router has a good multiple-beam performance, that is, sufficiently delivering wireless power to multiple targets while keeping the total RF-to-RF efficiency steady.

Figure. S33. (a) The double-target experimental setup. (b) The measured double-target RF-to-RF efficiency.

Figure. S34. (a) The triple-target experimental setup. (b) The measured triple-target RF-to-RF efficiency.

(2) DC-to-DC Efficiency

The measured DC-to-DC efficiencies at different distances and with different DC loads are shown in Figure. S35. As can be seen, as the input DC power P_{DCin} increases, the DC-to-DC efficiencies first rise and then fall. This phenomenon results from the saturation effect of the diode used in the rectifier, as shown in Figure. S30. With the increase of the input RF power, the output DC voltage of the rectifier also rises until it reaches the minimum breakdown voltage (MBV) of the diode. When the output DC

voltage surpasses the MBV, the diode operates in the saturation region and, in turn, lowers the entire WPT efficiency.

Figure. S35. The measured DC-to-DC efficiency. d is the distance between the router and the receiving antenna. R_{DC} is the DC load of the rectifier. (a) $d = 200$ mm; (b) $d = 400$ mm; (c) $d = 1000$ mm.

(3) Discussion on the WPT efficiency

The proposed WPT system has the best RF-to-RF of 3% and the best DC-to-DC efficiency (the entire WPT system) of 0.65%. Although the power transfer efficiency is on the order of several percent, it can still be comparable to recent state-of-the-art works [S3][S9]. Several reasons result in such an efficiency:

1) The diffraction-limited spot width. As the distance increases, the focal spot width increases according to the Abbe diffraction limitation by approximately

$$w = R \frac{\lambda}{D} \quad (6)$$

where R is the distance between the proposed router and the receiving antenna, λ is the wavelength, and D is the aperture size of the proposed router. On the other hand, the fixed receiving antenna's size results in the receiving antenna only capturing a partial portion of the delivered power. Diffraction-free beams that can propagate without diffractive spreading, such as Bessel beams, may improve the RF-to-RF efficiency. Another way to improve the efficiency can be increasing the aperture sizes of the router and the receiving antenna. The former can lead to smaller spot width according to Equation (6), while the latter enables capturing more delivered power.

2) The lossy substrates and diodes. The transmissive amplitudes of the RTM unit-cell are lower than unity due to the lossy substrates and diodes, as shown in Figure. S11.

The advance of material science that enables substrates and diodes with lower losses can alleviate this problem.

3) The lossy cables and connectors. These cables and connectors are used to connect the router with the signal generator and the receiving antenna with the rectifier. Although lossless cables and connectors may not be available in practice, shortening or eliminating them is possible by integrating these components (including cables, connectors, the signal generator, et al.) into a chip so as to reduce these losses."

Comment: 7. In the SWIPT experiment, the authors select space-time modulation rather than PSK or QAM modulation. The data rate is relatively low. Please consider using other modulations.

Response: We thank the reviewer for this comment. We would like to explain why we use the space-time modulation.

Firstly, the space-time modulation scheme allows to simultaneously realize beamforming and information modulation by the metasurface. To maintain compactness, the proposed router has no modulators or phase shifters. It only consists of a nearfield feeder (NFF), a metasurface, and two pluggable components (a signal generator and a control board). The beamforming is realized by changing the spatial phase distribution on the metasurface (that is, space modulation), while the information modulation is realized by changing the temporal transmissive phases (that is, time modulation). By combining the space and time modulation (space-time modulation), beamforming and information modulation are simultaneously realized by the metasurface.

Secondly, both practical PSK and QAM modulation schemes requires carrier synchronization, which may increase the cost and complexity of the router. Our manuscript focuses on a compact, low-cost WPT router rather than designing a complex receiving device. Designing a general receiver to handle various power-consuming devices is relatively difficult. In addition, the phase quantization of the proposed router is 2-bit, which limits the highest supported modulation order. Besides, QAM

modulation/demodulation scheme need two orthogonal carrier waves. This increases the hardware complexity of the proposed router and, thus, increases the potential cost.

Thirdly, for IoT, short packets with dozens of bits are transmitted. As a result, a low data rate, such as 50 baud, is adequate for these applications. In sensor networks, for example, short packet transmission usually includes environment parameters such as humidity, temperature, and atmospheric pressure, which require only a dozen or so bits to represent. A low data rate is sufficient for these scenarios.

It should be noted that the data rate is not mainly limited by the proposed router but by the receiver. The proposed router has a MHz modulation speed for information transmission. Nonetheless, the sampling time of the low-cost sampling chip used in the receiver is approximately 1 ms for each measurement. According to the Nyquist sampling theorem, the maximum allowed bandwidth of the transmitted signal is $1 / (1 \text{ ms}) / 2 = 500 \text{ Hz}$. In our demonstrations, the data rate is 50 baud. The corresponding symbol duration is 0.02 s. The bandwidth of the data can be estimated as around $1 / 0.02 * 2 = 100 \text{ Hz}$. Due to the signal tail and noise in practice, the actual maximum allowed bandwidth for the transmitted signal is smaller than 500 Hz, while the actual data bandwidth exceeds 100 Hz. Such a data rate is used to ensure that information is transmitted accurately and is also sufficient for short packet transmission in IoT scenarios.

We would like to explain how to improve the data rate while considering the proposed space-time modulation scheme. According to this scheme, we can map the transmitted symbol as follows: {Symbol#1 \rightarrow 0 Hz, Symbol#2 \rightarrow $(0 + f_0)$ Hz, Symbol#3 \rightarrow $(0 + 2f_0)$ Hz, ...}, where $f_0 > 0$ is dependent on the threshold of the symbol decision process. Accordingly, improving the data rate can be realized by 1) increasing the data bandwidth and 2) using better sampling chips to improve the sampling rate. These methods enable the transmission of more bits within a symbol duration.

We would like to express our gratitude to the reviewer for leading us to think about using other modulations. We would consider implementing other modulations in our future work.

Comment: 8. Grammatical errors and typos should be fixed.

Response: We thank the reviewer for this kind reminder. We have checked our manuscript carefully and corrected the typos.

REVIEWER COMMENTS

Reviewer #1 (Remarks to the Author):

I appreciate the effort of the authors in addressing all my previous concerns. I am satisfied with the current version. Thank you.

Reviewer #2 (Remarks to the Author):

The authors have sufficiently responded to my questions and comments. I have not further comments.

Reviewer #4 (Remarks to the Author):

The reviewer appreciates the authors' effort in revising their manuscript according to the reviewer's comments. However, some of the responses are not satisfied. Please consider the following comments:

1. In the response to comment 2 of Reviewer #4, the authors stated that they compared the profile of the proposed system with the other works, the thickness of the metasurface, the profile of the feeder, and the separation between the metasurface and the feeder, without considering the control board. Per the comment above, the complete router could not work without the control board. It does not make sense to readers if the authors stated the control board could be miniaturized by adopting customized integrated voltage chips, but not implemented in this work. Therefore, it is recommended not to claim the proposed reconfigurable power router with a low profile but the metasurface and feeder only to avoid confusion.
2. Regarding comment 3 of Reviewer #4, AMC is a metallic array printed on a ground dielectric substrate that can fully reflect incident waves with a near-zero reflection phase. What is the difference between the AMC and a normal metal panel since a metal panel can fully reflect the incident wave, and the reflection phase depends on the distance between the investigated point and the structure (e.g., AMC or metal panel)? If there is any particular characteristic, please clarify it.
3. In response to comment 4 of Reviewer #4, please clearly state the exact transmit power with the corresponding test scenarios in the manuscript. Please provide the information about the signal generator (e.g., model name, part number) that can deliver 50W output power. If the authors used a high-power RF amplifier, please provide the information. Furthermore, 50W is exceptionally high for a practical router. This extremely high transmit power required in the test comes from immensely low RF-to-RF and DC-to-DC power transfer efficiency (PTE) mentioned in the response. This low PTE makes the proposed system not suitable for practical scenarios.
4. In response to comment 5 of Reviewer #4, the Reviewer agrees that it is possible to charge daily-life electronic devices with 5 V. However, with extremely low PTE due to using only one rectenna, as given in Figure S41 of Supplementary Note 11, the harvesting current is believed to be tremendously low to charge a smartphone or a power bank, or it will take forever to be charged when it comes to practice. It is suggested to incorporate multiple rectennas to increase the harvested power, as demonstrated in [R1].

[R1] N. M. Tran, M. M. Amri, J. H. Park, D. I. Kim and K. W. Choi, "Reconfigurable-Intelligent-Surface-Aided Wireless Power Transfer Systems: Analysis and Implementation," in IEEE Internet of Things Journal, vol. 9, no. 21, pp. 21338-21356, 1 Nov.1, 2022, doi: 10.1109/JIOT.2022.3179691.

5. In response to comment 6 of Reviewer #4, though the proposed system utilizes a beamforming method to form a beam toward the receivers, the RF-to-RF efficiency is too low compared to existing works [R2-R3]. For example, as demonstrated in [R2], the PTE is around 27% at 400 mm at 5.8 GHz operating frequency. In addition, the PTE is much higher, with around 45%, at 400 mm in [R3]. Meanwhile, the maximum RF-to-RF efficiency in the proposed system is just around 3% and the maximum DC-to-DC efficiency is just about 0.65% at 200 mm. Though the proposed system has a compact size and low profile, the performance is not good enough. Could the authors comment on this matter?

[R2] J. Han et al., "Adaptively Smart Wireless Power Transfer Using 2-Bit Programmable Metasurface," in IEEE Transactions on Industrial Electronics, vol. 69, no. 8, pp. 8524-8534, Aug. 2022, doi: 10.1109/TIE.2021.

[R3] 3105988J. H. Park, D. I. Kim and K. W. Choi, "Analysis and Experiment on Multi-Antenna-to-Multi-Antenna RF Wireless Power Transfer," in IEEE Access, vol. 9, pp. 2018-2031, 2021, doi: 10.1109/ACCESS.2020.3047485.

6. Furthermore, as mentioned by the authors, the maximum transmit power generated by the tunable signal generator in the experiment is 50 W. How could authors investigate the DC-to-DC efficiency with up to 90 W of P_{DCin} as displayed in Figure S35? Please clarify it.

7. The main contribution of this work is designing a novel structure of NFF to generate a uniform excitation that feeds to the DBG. However, since authors can control the transmission phase of the 2-bit RTM in DBG, the DBG can form one or multiple power beams with a spherical wave source excitation when a proper transmission phase is set at the BDG to compensate for the non-uniform phase. Could the authors give comments on it?

Reviewer #1's comments on Reviewer #3's remaining concerns (response to questions posed by editor based on Reviewer #3's previous review report):

1. Do you feel that the introduction contains sufficient background information on self-adaptive metasurfaces?
2. Is the design procedure for the metasurface unit cell in Supplementary Note 3 adequately presented?
3. Does Supplementary Note 7 provide a sufficient characterisation of the rectifier design?

I think the authors did address them successfully. My only recommendation for the authors is to proofread once more the paper as sometimes the ideas are not crystal clear. Some claims may not be even formulated correctly, e.g., in Supplementary Note 8, (3) Discussion on the WPT Efficiency, the statement: "Although the power transfer efficiency is on the order of several percent,..." may not be correct since the WPT efficiency was shown to be 0.65% in this case...

Reviewer #4's comments on Reviewer #3's remaining concerns (response to questions posed by editor based on Reviewer #3's previous review report):

1. Do you feel that the introduction contains sufficient background information on self-adaptive metasurfaces?

In Reviewer's perspective, self-adaptive metasurfaces share the same idea concept with programmable/coding metasurfaces, reconfigurable intelligent surface, intelligent reflecting surface. The background information on self-adaptive metasurfaces is not adequate, however, it is acceptable. It is suggested that authors should clearly emphasize what kind of metasurfaces are to be proposed and give more information and references for that metasurface's concept.

2. Is the design procedure for the metasurface unit cell in Supplementary Note 3 adequately presented?

The design of the metasurface unit cell in this work might be inspired by the work [62] and [64] in the main text (elliptical u-slot radiator in [62], additional delay line and PIN diodes for 2-bit transmission phase in [64]).

The design procedure and operation principle of the proposed unit cell seems not much different to that presented in [64]. Please request authors to highlight the novelty of the proposed design compared to the above-mentioned references.

[62] Bai, X. et al. High-efficiency transmissive programmable metasurface for multimode OAM generation. Adv. Opt. Mater. 8, 2000570 (2020).

[64] Diaby, F. et al. 2-bit reconfigurable unit-cell and electronically steerable transmitarray at 18 Ka-band. *IEEE Trans. Antennas Propag.* 68, 5003-5008 (2019).

3. Does Supplementary Note 7 provide a sufficient characterisation of the rectifier design?

Since rectifier design is not the main contribution of this paper, the full description of the rectifier design can be excluded. There are several efficient rectifier topologies (e.g., full wave rectifier, Dickson charge pump), however, authors have chosen the most basic rectifier topology (i.e., half wave rectifier) in the paper. This topology may lower the RF-to-DC conversion efficiency. Could authors explain the reason for selecting the proposed design?

Furthermore, the equation for calculating the rectifier efficiency in Supplement Note 7 is wrong. Authors should correct it. In figure S30(c), what is the input power for all frequency bands? Does the rectifier give the maximum efficiency over all frequencies with the same input power? Please ask authors to clarify it.

Reviewer #1 (Remarks to the Author):

I appreciate the effort of the authors in addressing all my previous concerns. I am satisfied with the current version. Thank you.

Reviewer #2 (Remarks to the Author):

The authors have sufficiently responded to my questions and comments. I have not further comments.

Reviewer #4 (Remarks to the Author):

The reviewer appreciates the authors' effort in revising their manuscript according to the reviewer's comments. However, some of the responses are not satisfied. Please consider the following comments:

1. In the response to comment 2 of Reviewer #4, the authors stated that they compared the profile of the proposed system with the other works, the thickness of the metasurface, the profile of the feeder, and the separation between the metasurface and the feeder, without considering the control board. Per the comment above, the complete router could not work without the control board. It does not make sense to readers if the authors stated the control board could be miniaturized by adopting customized integrated voltage chips, but not implemented in this work. Therefore, it is recommended not to claim the proposed reconfigurable power router with a low profile but the metasurface and feeder only to avoid confusion.
2. Regarding comment 3 of Reviewer #4, AMC is a metallic array printed on a ground dielectric substrate that can fully reflect incident waves with a near-zero reflection phase. What is the difference between the AMC and a normal metal panel since a metal panel can fully reflect the incident wave, and the reflection phase depends on the distance between the investigated point and the structure (e.g., AMC or metal panel)? If there is any particular characteristic, please clarify it.
3. In response to comment 4 of Reviewer #4, please clearly state the exact transmit power with the corresponding test scenarios in the manuscript. Please provide the information about the signal generator (e.g., model name, part number) that can deliver 50W output power. If the authors used a high-power RF amplifier, please provide the information. Furthermore, 50W is exceptionally high for a practical router. This extremely high transmit power required in the test comes from immensely low RF-to-RF and DC-to-DC power transfer efficiency (PTE) mentioned in the response. This low PTE makes the proposed system not suitable for practical scenarios.
4. In response to comment 5 of Reviewer #4, the Reviewer agrees that it is possible to charge daily-life electronic devices with 5 V. However, with extremely low PTE due to using only one rectenna, as given in Figure S41 of Supplementary Note 11, the harvesting current is believed to be tremendously low to charge a smartphone or a power bank, or it will take forever to be

charged when it comes to practice. It is suggested to incorporate multiple rectennas to increase the harvested power, as demonstrated in [R1].

[R1] N. M. Tran, M. M. Amri, J. H. Park, D. I. Kim and K. W. Choi, "Reconfigurable-Intelligent-Surface-Aided Wireless Power Transfer Systems: Analysis and Implementation," in IEEE Internet of Things Journal, vol. 9, no. 21, pp. 21338-21356, 1 Nov.1, 2022, doi: 10.1109/JIOT.2022.3179691.

5. In response to comment 6 of Reviewer #4, though the proposed system utilizes a beamforming method to form a beam toward the receivers, the RF-to-RF efficiency is too low compared to existing works [R2-R3]. For example, as demonstrated in [R2], the PTE is around 27% at 400 mm at 5.8 GHz operating frequency. In addition, the PTE is much higher, with around 45%, at 400 mm in [R3]. Meanwhile, the maximum RF-to-RF efficiency in the proposed system is just around 3% and the maximum DC-to-DC efficiency is just about 0.65% at 200 mm. Though the proposed system has a compact size and low profile, the performance is not good enough. Could the authors comment on this matter?

[R2] J. Han et al., "Adaptively Smart Wireless Power Transfer Using 2-Bit Programmable Metasurface," in IEEE Transactions on Industrial Electronics, vol. 69, no. 8, pp. 8524-8534, Aug. 2022, doi: 10.1109/TIE.2021.

[R3] 3105988]. H. Park, D. I. Kim and K. W. Choi, "Analysis and Experiment on Multi-Antenna-to-Multi-Antenna RF Wireless Power Transfer," in IEEE Access, vol. 9, pp. 2018-2031, 2021, doi: 10.1109/ACCESS.2020.3047485.

6. Furthermore, as mentioned by the authors, the maximum transmit power generated by the tunable signal generator in the experiment is 50 W. How could authors investigate the DC-to-DC efficiency with up to 90 W of P_{DCin} as displayed in Figure S35? Please clarify it.

7. The main contribution of this work is designing a novel structure of NFF to generate a uniform excitation that feeds to the DBG. However, since authors can control the transmission phase of the 2-bit RTM in DBG, the DBG can form one or multiple power beams with a spherical wave source excitation when a proper transmission phase is set at the BDG to compensate for the non-uniform phase. Could the authors give comments on it?

Reviewer #1's comments on Reviewer #3's remaining concerns (response to questions posed by editor based on Reviewer #3's previous review report):

1. Do you feel that the introduction contains sufficient background information on self-adaptive metasurfaces?
2. Is the design procedure for the metasurface unit cell in Supplementary Note 3 adequately presented?
3. Does Supplementary Note 7 provide a sufficient characterisation of the rectifier design?

I think the authors did address them successfully. My only recommendation for the authors is to proofread once more the paper as sometimes the ideas are not crystal clear. Some claims may not be even formulated correctly, e.g., in Supplementary Note 8, (3) Discussion on the WPT Efficiency, the statement: "Although the power transfer efficiency is on the order of several percent,..." may not be correct since the WPT efficiency was shown to be 0.65% in this case...

Reviewer #4's comments on Reviewer #3's remaining concerns (response to questions posed by editor based on Reviewer #3's previous review report):

1. Do you feel that the introduction contains sufficient background information on self-adaptive metasurfaces?

In Reviewer's perspective, self-adaptive metasurfaces share the same idea concept with programmable/coding metasurfaces, reconfigurable intelligent surface, intelligent reflecting surface. The background information on self-adaptive metasurfaces is not adequate, however, it is acceptable. It is suggested that authors should clearly emphasize what kind of metasurfaces are to be proposed and give more information and references for that metasurface's concept.

2. Is the design procedure for the metasurface unit cell in Supplementary Note 3 adequately presented?

The design of the metasurface unit cell in this work might be inspired by the work [62] and [64] in the main text (elliptical u-slot radiator in [62], additional delay line and PIN diodes for 2-bit transmission phase in [64]).

The design procedure and operation principle of the proposed unit cell seems not much different to that presented in [64]. Please request authors to highlight the novelty of the proposed design compared to the above-mentioned references.

[62] Bai, X. et al. High-efficiency transmissive programmable metasurface for multimode OAM generation. *Adv. Opt. Mater.* 8, 2000570 (2020).

[64] Diaby, F. et al. 2-bit reconfigurable unit-cell and electronically steerable transmitarray at 18 Ka-band. *IEEE Trans. Antennas Propag.* 68, 5003-5008 (2019).

3. Does Supplementary Note 7 provide a sufficient characterisation of the rectifier design?

Since rectifier design is not the main contribution of this paper, the full description of the rectifier design can be excluded. There are several efficient rectifier topologies (e.g., full wave rectifier, Dickson charge pump), however, authors have chosen the most basic rectifier topology (i.e., half wave rectifier) in the paper. This topology may lower the RF-to-DC conversion efficiency. Could authors explain the reason for selecting the proposed design?

Furthermore, the equation for calculating the rectifier efficiency in Supplement Note 7 is wrong. Authors should correct it. In figure S30(c), what is the input power for all frequency

bands? Does the rectifier give the maximum efficiency over all frequencies with the same input power? Please ask authors to clarify it.

Response to the Reviewers' Comments: We would like to express our gratitude to the editor and reviewers for appreciation and constructive comments on our manuscript. We are also thankful for their inspiring comments which have allowed us to greatly improve the manuscript. We have followed the reviewers' advices and carefully modified the manuscript. Hopefully, our reply and the corresponding revision made to the manuscript can meet all requirements of the reviewers. Our detailed replies are listed as follows.

Responses to Reviewer #1:

Comment: Reviewer #1 (Remarks to the Author):

I appreciate the effort of the authors in addressing all my previous concerns. I am satisfied with the current version. Thank you.

Response: We truly thank the reviewer for the positive comment.

Comment: Reviewer #1's comments on Reviewer #3's remaining concerns (response to questions posed by editor based on Reviewer #3's previous review report):

1. Do you feel that the introduction contains sufficient background information on self-adaptive metasurfaces?
2. Is the design procedure for the metasurface unit cell in Supplementary Note 3 adequately presented?
3. Does Supplementary Note 7 provide a sufficient characterisation of the rectifier design?

I think the authors did address them successfully. My only recommendation for the authors is to proofread once more the paper as sometimes the ideas are not crystal clear. Some claims may not be even formulated correctly, e.g., in Supplementary Note 8, (3) Discussion on the WPT Efficiency, the statement: "Although the power transfer efficiency is on the order of several percent,..." may not be correct since the WPT efficiency was shown to be 0.65% in this case...

Response: We thank the reviewer for the positive comment. Following the reviewer's helpful comments, we have carefully revised the entire manuscript.

Revised Text:

Lines 1-2 on Page 32 of Supplementary Note 8:

"The proposed WPT system has the best RF-to-RF of 3% and the best DC-to-DC efficiency (the entire WPT system) of 0.65%. The RF-to-RF efficiency is on the order

of several percent, which is comparable to recent works [S3][S9]. "

Responses to Reviewer #2:

Comment: Reviewer #2 (Remarks to the Author):

The authors have sufficiently responded to my questions and comments. I have not further comments.

Response: We truly appreciate the reviewer's positive comment.

Responses to Reviewer #4:

Comment: Reviewer #4 (Remarks to the Author):

The reviewer appreciates the authors' effort in revising their manuscript according to the reviewer's comments. However, some of the responses are not satisfied. Please consider the following comments:

Response: We appreciate the reviewer's valuable comments. The insightful comments are very helpful in further refining our manuscript. In the following, we address the specific comments point-by-point while revising our manuscript.

Comment: 1. In the response to comment 2 of Reviewer #4, the authors stated that they compared the profile of the proposed system with the other works, the thickness of the metasurface, the profile of the feeder, and the separation between the metasurface and the feeder, without considering the control board. Per the comment above, the complete router could not work without the control board. It does not make sense to readers if the authors stated the control board could be miniaturized by adopting customized integrated voltage chips, but not implemented in this work. Therefore, it is recommended not to claim the proposed reconfigurable power router with a low profile but the metasurface and feeder only to avoid confusion.

Response: We thank the reviewer for this comment. We have modified the original statement about the profile to avoid confusion. It is clearly stated that the metasurface and the feeder have a combined profile of 0.8λ . We have revised the corresponding statements in main text. Please see lines 19–20 on page 1 and lines 4–5 on page 4 of the main text.

Revised Text:

In the main text:

"The combined profile of the feeder and the beam generator is 0.8λ ."

"The total profile of the two components is 0.8 wavelength."

Comment: 2. Regarding comment 3 of Reviewer #4, AMC is a metallic array printed on a ground dielectric substrate that can fully reflect incident waves with a near-zero reflection phase. What is the difference between the AMC and a normal metal panel since a metal panel can fully reflect the incident wave, and the reflection phase depends on the distance between the investigated point and the structure (e.g., AMC or metal panel)? If there is any particular characteristic, please clarify it.

Response: We thank the reviewer for this comment. To maintain a low profile, the AMC or metal panel is usually placed close to the investigated point (or the source), as shown in Figure. R4 - 1. The resonant condition can be written as [R1]

$$h = \frac{\varphi_a + \varphi_p}{4\pi} \lambda + N \frac{\lambda}{2} \quad (1)$$

where h is the height of the nearfield feeder (that is, the separation distance between the AMC (or the metal plane) and the PRP), φ_a is the reflective phase of the AMC or the metal panel, φ_p is the reflective phase of the PRP, λ is the working wavelength, and $N = 0, 1, 2, \dots$.

In our design, the reflection phase of the PRP is close to π , as shown in Figure 2b (I) of the main text. In Figure. R4 - 1(a), since the reflection phase of a metal panel is π , the height of the nearfield feeder is close to $\lambda/2$ with $N = 0$. In Figure. R4 - 1(b), the reflection phase of AMC is close to 0 (as shown in Figure 2b (II) of the main text), and thus, the height of the nearfield feeder is close to $\lambda/4$ with $N = 0$, which is smaller than that in Figure. R4 - 1(a). In Figure. R4 - 1(c), the reflective phase φ_a is tuned to $0 + 2M\pi$ by moving the metal panel away from the source. By considering the roundtrip in the substrate and the reflective phase of a metal panel, the reflective phase φ_a is $4\pi h_1 \sqrt{\varepsilon_r} / \lambda + \pi$, where ε_r is the dielectric constant of the substrate. We can obtain the thickness of the substrate is $h_1 = \lambda / (4\sqrt{\varepsilon_r})$ with $M = 1$. The total height of the nearfield feeder is $h \approx \lambda / 4 + \lambda / (4\sqrt{\varepsilon_r}) > \lambda / 4$. Given the operating wavelength $\lambda = 51.2$ mm and the dielectric constant $\varepsilon_r = 2.2$ in our manuscript, the heights of the nearfield feeders in Figure. R4 - 1 (a), (b), and (c) are 25.85 mm, 12.93 mm, and 21.64 mm, respectively. Although the φ_a in Figure. R4 - 1(c) can be tuned by moving the metal panel, it requires

increasing the distance between the metal panel and the source, which in turn increases the profile of the nearfield feeder. Therefore, the use of the AMC enables a lower profile compared with the metal panel.

Figure. R4 - 1. Nearfield feeder formed by (a) a metal panel and PRP, (b) AMC and PRP. (c) The near-zero phase is achieved by moving the metal panel. PRP: partial reflective plane.

[R1] Kelly J R, Kokkinos T, Feresidis A P. Analysis and design of sub-wavelength resonant cavity type 2-D leaky-wave antennas. IEEE transactions on antennas and propagation, 2008, 56(9): 2817-2825.

Comment: 3. In response to comment 4 of Reviewer #4, please clearly state the exact transmit power with the corresponding test scenarios in the manuscript. Please provide the information about the signal generator (e.g., model name, part number) that can deliver 50W output power. If the authors used a high-power RF amplifier, please provide the information. Furthermore, 50W is exceptionally high for a practical router. This extremely high transmit power required in the test comes from immensely low RF-to-RF and DC-to-DC power transfer efficiency (PTE) mentioned in the response. This low PTE makes the proposed system not suitable for practical scenarios.

Response: We thank the reviewer for this comment. In the experiment for lighting up the LED, the transmitted power, i.e., the output RF power of the tunable signal generator, is 8 W. In the experiment for charging the smartphone and the power bank, the transmitted power is 50 W. We have clearly stated the transmitted power in the revised manuscript. Please see lines 4 and 24 on page 10 of the main text.

The signal generator (WPS-5800-50W) is commercially available from Chendu Jiaxi Science and Technology Limited Liability Company, China. Figure. R4 - 2 shows the block diagram of the signal generator. The power conversion module converts the 220V AC input (the standard household voltage in the author's country) to DC power to drive all other components. The RF signal is generated by the signal source and then amplified by the driving amplifier and the power amplifiers. The RF output power level is controlled by the control and protection module. We have supplemented the information about the signal generator in Supplementary Note 8. Please see lines 12–19 on page 30 and Figure S35 on page 31 of Supplementary Note.

Figure. R4 - 2. The block diagram of the signal generator.

The 50 W RF power is not used in all our demonstrations. In the experiment, we use the 50W RF power to charge the smartphone and the power bank, since at this power level, the output voltage of the rectifier reaches 5 V for these daily-life electronic devices. In this way, we can experimentally test the feasibility of the proposed system in practical scenarios.

The main source of the received power bottleneck can be attributed to the relatively small receiver size (50 mm × 30 mm) and rectifier efficiency. Therefore, there are two methods to improve the PTE: 1) incorporating multiple receiving antennas to improve the receiving gain; and 2) using better rectifying diodes to improve the RF-to-DC

efficiency. A representatively successful design can be referred to in Ref. [66][67] of the main text. We have supplemented the corresponding discussion in the main text. Please see lines 29–31 on page 11 of the main text.

Revised Text:

In the main text:

"In this experiment, the input RF power to the router is 8 W."

"The router's input RF power is 50 W in this case."

"An important way to improve the power transfer efficiency is by incorporating multiple receiving antennas [65][66]. This can increase the receiving gain, enabling charging devices with higher power requirements."

In Supplementary Note 8:

"We use a tunable signal generator to test the DC-to-DC efficiency. The signal generator (WPS-5800-50W) is commercially available from Chendu Jiaxi Science and Technology Limited Liability Company, China. It has a DC-to-RF efficiency of 40%. The block diagram of the signal generator is shown in Figure. S35. The power conversion module converts the 220V AC input (the standard household voltage in the author's country) to DC power to drive all other components. The RF signal is generated by the signal source and then amplified by the driving amplifier and the power amplifiers. The RF output power level is controlled by the control and protection module."

Comment: 4. In response to comment 5 of Reviewer #4, the Reviewer agrees that it is possible to charge daily-life electronic devices with 5 V. However, with extremely low PTE due to using only one rectenna, as given in Figure S41 of Supplementary Note 11, the harvesting current is believed to be tremendously low to charge a smartphone or a power bank, or it will take forever to be charged when it comes to practice. It is suggested to incorporate multiple rectennas to increase the harvested power, as demonstrated in [R1].

[R1] N. M. Tran, M. M. Amri, J. H. Park, D. I. Kim and K. W. Choi, "Reconfigurable-Intelligent-Surface-Aided Wireless Power Transfer Systems: Analysis and Implementation," in IEEE Internet of Things Journal, vol. 9, no. 21, pp. 21338-21356, 1 Nov.1, 2022, doi: 10.1109/JIOT.2022.3179691.

Response: We thank the reviewer for this comment. We agree that the PTE is not high for charging the smartphone or the power bank due to the adoption of a single rectenna. Incorporating multiple rectennas can be an effective way to increase the harvested power for devices with higher power requirements. To enhance this point, we have supplemented relevant discussion. Please see lines 29–31 on page 11 of the main text.

Revised Text:

In the main text:

"An important way to improve the power transfer efficiency is by incorporating multiple receiving antennas [65][66]. This can increase the receiving gain, enabling charging devices with higher power requirements."

Comment: 5. In response to comment 6 of Reviewer #4, though the proposed system utilizes a beamforming method to form a beam toward the receivers, the RF-to-RF efficiency is too low compared to existing works [R2-R3]. For example, as demonstrated in [R2], the PTE is around 27% at 400 mm at 5.8 GHz operating frequency. In addition, the PTE is much higher, with around 45%, at 400 mm in [R3]. Meanwhile, the maximum RF-to-RF efficiency in the proposed system is just around 3% and the maximum DC-to-DC efficiency is just about 0.65% at 200 mm. Though the proposed system has a compact size and low profile, the performance is not good enough. Could the authors comment on this matter?

[R2] J. Han et al., "Adaptively Smart Wireless Power Transfer Using 2-Bit Programmable Metasurface," in IEEE Transactions on Industrial Electronics, vol. 69, no. 8, pp. 8524-8534, Aug. 2022, doi: 10.1109/TIE.2021.3105988

[R3] J. H. Park, D. I. Kim and K. W. Choi, "Analysis and Experiment on Multi-

Antenna-to-Multi-Antenna RF Wireless Power Transfer," in IEEE Access, vol. 9, pp. 2018-2031, 2021, doi: 10.1109/ACCESS.2020.3047485.

Response: We thank the reviewer for this comment. The higher PTE in [R2] and [R3] may be attributed to the following reasons:

1) Larger receivers are used in [R2] and [R3]. The receiver sizes are **120 mm × 120 mm** in [R2] and **113 mm × 113 mm** in [R3]. In our work, the receiver size is **50 mm × 30 mm**, which is selected for small devices and easy-to-integrate uses. The sizes in [R2] and [R3] are **9.6** and **8.5** times greater than that in our work, respectively, leading to higher receiving antenna gains. The larger receiver sizes could be the main contribution to the higher PTE in [R2] and [R3].

2) In [R2], the unit-cell is reflective and has less substrate layers and less diodes, leading to lower substrate and diode losses. In [R2], the unit-cell consists of 3 substrate layers and 2 PIN diodes, as shown in Figure 5 in [R2]. In our work, the unit-cell is a transmissive structure and consists of 9 substrate layers and 4 PIN diodes, as shown in Figure 3a of the main text. This can result in higher losses compared with [R2].

3) In [R3], phased arrays with 6-bit digital phase shifters (64 phase levels) are used, which provide higher phase resolution for manipulating waves. In our work, we use 4 PIN diodes to achieve 2-bit phase resolution (4 phase levels). The higher phase resolution can also contribute to the higher PTE.

Comment: 6. Furthermore, as mentioned by the authors, the maximum transmit power generated by the tunable signal generator in the experiment is 50 W. How could authors investigate the DC-to-DC efficiency with up to 90 W of P_{DCin} as displayed in Figure S35? Please clarify it.

Response: We thank the reviewer for this comment. P_{DCin} is the input DC power to the signal generator, which has a DC-to-RF efficiency of 40%. As the output RF power ranges from 30 to 45 dBm, the corresponding DC power ranges from around 2.5 to 90 W. The P_{DCin} in Figure S35 corresponds to this DC power range. We have supplemented the corresponding description in Supplementary Note 8. Please see lines 4–5 on page

31 of Supplementary Note.

Revised Text:

In Supplementary Note 8:

"The input DC power to the signal generator ranges from around 2.5 W to 90 W (the corresponding output RF power ranges from 30 dBm to 45 dBm)."

Comment: 7. The main contribution of this work is designing a novel structure of NFF to generate a uniform excitation that feeds to the DBG. However, since authors can control the transmission phase of the 2-bit RTM in DBG, the DBG can form one or multiple power beams with a spherical wave source excitation when a proper transmission phase is set at the BDG to compensate for the non-uniform phase. Could the authors give comments on it?

Response: We thank the reviewer for this comment. We agree that the DBG can form one or multiple beams with spherical wave source excitations. However, spherical wave sources (such as horn antennas, open-end rectangle waveguide antennas) can result in large system profiles. Figure. R4 - 3 shows how a spherical wave source excites a metasurface. On one hand, to illuminate all unit-cells in the metasurface, the spherical wave source requires a large distance away from the metasurface, as shown in Figure. R4 - 3(a). In other words, there needs a relatively significant distance between the spherical wave source and the metasurface to ensure that the beam spreads widely enough to excite the metasurface uniformly. On the other hand, when the distance between the spherical wave source and the metasurface is reduced, the metasurface cannot be sufficiently excited due to the 'non-spreading' of the spherical waves, as shown in Figure. R4 - 3(b).

Figure. R4 - 3. (a) Large profile. (b) Small profile.

We would like to give a qualitative picture to explain the working mechanism of the nearfield feeder (NFF). The proposed NFF are composed of a 2×2 antenna array, an artificial magnetic conductor (AMC) and a partial reflective plane (PRP). The latter two components create a Fabry-Perot cavity (FPC). The powers are injected into the FPC through the antenna array. Due to the highly reflective properties of the AMC and the PRP, the injected powers first spread through the FPC (going back and forth between the AMC and the PRP). Arriving at the PRP, the injected powers partially leak out due to the partial reflection property of the PRP. As this process repeats, a uniform wavefront is formed in the nearfield of the NFF, as shown in Figure. R4 - 4. Such a nearfield distribution can be directly fed to the metasurface to acquire uniform excitation. In this way, the NFF not only excites all unit-cells properly but also maintains the low profile.

Figure. R4 - 4. The qualitative explanation of the NFF. From left to right, the first two layers are AMC and PRP, respectively, while the third layer is the metasurface.

Comment: Reviewer #4's comments on Reviewer #3's remaining concerns (response to questions posed by editor based on Reviewer #3's previous review report):

1. Do you feel that the introduction contains sufficient background information on self-adaptive metasurfaces?

In Reviewer's perspective, self-adaptive metasurfaces share the same idea concept with programmable/coding metasurfaces, reconfigurable intelligent surface, intelligent reflecting surface. The background information on self-adaptive metasurfaces is not adequate, however, it is acceptable. It is suggested that authors should clearly emphasize what kind of metasurfaces are to be proposed and give more information and references for that metasurface's concept.

Response: We thank the reviewer for this comment. The proposed metasurface is a programmable metasurface with space-time modulation. Programmable metasurfaces have been studied over the past decade. However, the potential advantages of metasurfaces cannot be fully realized due to their time/frequency-invariant responses. To overcome these, space-time-modulated programmable metasurfaces are introduced for simultaneous manipulations in spatial and frequency domains [R1][R2]. Due to their remarkable properties, space-time-modulated programmable metasurface are used for

many applications, such as nonreciprocal transmission, frequency conversion, and wave amplification.

We have emphasized that the dynamic beam generator (DBG) can be categorized as what kind of metasurface and provided the corresponding references to this kind of metasurface. Please see lines 6–11 on page 3, lines 28–30 on page 4, and Ref. [35]-[39] of the main text.

Revised Text:

In the main text:

"Another area of significant interest is programmable metasurfaces [35], which utilize digitally programmable unit-cells to provide a link between the physical and digital worlds [36]. Thanks to their prominent characteristics, programmable metasurfaces have found applications in various domains, including space-time modulations [37][38][39], polarization controls [40], and meta-imagers [41][42]."

"The DBG is a space-time-modulated programmable metasurface that consists of many active transmissive unit-cells."

[R1] Hu Q, Yang W, Wang J, et al. Dynamically Generating Diverse Multi-Beams with On-Demand Polarizations through Space-Time Coding Metasurface. *Advanced Optical Materials*, 2023: 2300093.

[R2] Wu G B, Dai J Y, Cheng Q, et al. Sideband-free space–time-coding metasurface antennas. *Nature electronics*, 2022, 5(11): 808-819.

Comment: 2. Is the design procedure for the metasurface unit cell in Supplementary Note 3 adequately presented?

The design of the metasurface unit cell in this work might be inspired by the work [62] and [64] in the main text (elliptical u-slot radiator in [62], additional delay line and PIN diodes for 2-bit transmission phase in [64]).

The design procedure and operation principle of the proposed unit cell seems not much different to that presented in [64]. Please request authors to highlight the novelty of the

proposed design compared to the above-mentioned references.

[62] Bai, X. et al. High-efficiency transmissive programmable metasurface for multimode OAM generation. *Adv. Opt. Mater.* 8, 2000570 (2020).

[64] Diaby, F. et al. 2-bit reconfigurable unit-cell and electronically steerable transmitarray at 18 Ka-band. *IEEE Trans. Antennas Propag.* 68, 5003-5008 (2019).

Response: We thank the reviewer for this comment. In Refs. [62] and [64], only a single ground is used, resulting in the DC line and the DC layer being placed in the same substrate layer, as shown in Figure. R4 - 5(a). Since the DC line and the DC share the same substrate layer, the coupling between the DC layer and the DC lines from other unit-cells may deteriorate the metasurface performance. Besides, the DC layer and the DC lines compete for the same space, limiting the placement of more DC lines from other unit-cells. This can bring difficulty to the fabrication of metasurfaces with larger sizes. To address these constraints, we introduce a double-ground structure to the proposed unit-cell, as shown in Figure 3a of the main text. All DC lines are 'hidden' between the two ground. The double-ground design can sufficiently avoid the coupling between the DC lines and the DC layer since they are placed on opposite sides of the ground, as shown in Figure. R4 - 5(b). Moreover, the two-ground design makes the fabrication of larger-size metasurfaces easier since the DC lines and the DC layer do not share the same space.

Figure. R4 - 5. (a) Single-ground design. (b) Double-ground design.

Comment: 3. Does Supplementary Note 7 provide a sufficient characterisation of the rectifier design?

Since rectifier design is not the main contribution of this paper, the full description of the rectifier design can be excluded. There are several efficient rectifier topologies (e.g., full wave rectifier, Dickson charge pump), however, authors have chosen the most basic rectifier topology (i.e., half wave rectifier) in the paper. This topology may lower the RF-to-DC conversion efficiency. Could authors explain the reason for selecting the proposed design?

Furthermore, the equation for calculating the rectifier efficiency in Supplement Note 7 is wrong. Authors should correct it. In figure S30(c), what is the input power for all frequency bands? Does the rectifier give the maximum efficiency over all frequencies with the same input power? Please ask authors to clarify it.

Response: We thank the reviewer for this comment. The proposed rectifier has a relatively simple structure and a small size of 3 cm × 2 cm, as shown in Figure S30(a) of Supplementary Note 7. It contains only one lumped element, i.e., a diode. This makes the rectifier low-cost and easy to use. Such a rectifier is selected for potential use in small-size devices, such as temperature and humidity sensors. As the reviewer points out, the rectifier design is not the main focus of our manuscript. We selected this rectifier to demonstrate the proposed router's capabilities for dynamically tracking and charging multiple devices.

We have corrected the equation for calculating the rectifier efficiency. In Figure S30(c) of Supplementary Note 7, the efficiencies are measured with the same input power of 20 dBm. The maximum efficiency in Figure S30(c) of Supplementary Note 7 refers to the RF-to-DC efficiency reaching its maximum within the input power range (-5 dBm to 20 dBm). The measured efficiencies over all frequencies are supplemented in Figure S30(b) of Supplementary Note 7. All curves in Figure S30(b) of Supplementary Note 7 have a similar tendency and reach their maximum at an input power of 20 dBm. We have supplemented the corresponding description in Supplementary Note 7. Please see lines 6–12 on page 27 of Supplementary Note.

Revised Text:

In Supplementary Note 7:

"The rectifier efficiency is defined as $\eta = P_{DC} / P_{in}$, where P_{DC} is the output DC power

and P_{in} is the input RF power.

Figure. S30. (a) The detailed dimensions of the rectifier. (b) The measured RF-to-DC efficiency at 5.6 GHz, 5.7 GHz, 5.8 GHz, 5.9 GHz, and 6.0 GHz; (c) The measured efficiency over frequency with an input power of 20 dBm."

REVIEWER COMMENTS

Reviewer #4 (Remarks to the Author):

I appreciate the effort of the authors in addressing the Reviewers' comments. I am satisfied with almost all the responses. There is one remaining concern as below:

In response to Reviewer #4's comments on Reviewer #3's remaining concerns, "Comment 2", the author has highlighted the novelty of the proposed work is the "double ground design" compared to the "single ground design" of the others. It is claimed by the authors that "the coupling between the DC layer and the DC lines from other unit-cells may deteriorate the metasurface performance." Please provide evidence (e.g., simulation results) for the above claim.

Thank you!

Reviewer #4 (Remarks to the Author):

I appreciate the effort of the authors in addressing the Reviewers' comments. I am satisfied with almost all the responses. There is one remaining concern as below:

In response to Reviewer #4's comments on Reviewer #3's remaining concerns, "Comment 2", the author has highlighted the novelty of the proposed work is the "double ground design" compared to the "single ground design" of the others. It is claimed by the authors that "the coupling between the DC layer and the DC lines from other unit-cells may deteriorate the metasurface performance." Please provide evidence (e.g., simulation results) for the above claim.

Thank you!

Response to the Reviewers' Comments: We would like to express our gratitude to the editor and Reviewer #4 for appreciation and constructive comments on our manuscript. We are also thankful for their inspiring comments which have allowed us to greatly improve the manuscript. Hopefully, our reply and the corresponding revision made to the manuscript can meet all requirements of Reviewer #4. Our detailed replies are listed as follows.

Responses to Reviewer #4:

Comment: Reviewer #4 (Remarks to the Author):

I appreciate the effort of the authors in addressing the Reviewers' comments. I am satisfied with almost all the responses. There is one remaining concern as below:

Response: We thank the reviewer for the comment.

Comment: In response to Reviewer #4's comments on Reviewer #3's remaining concerns, "Comment 2", the author has highlighted the novelty of the proposed work is the "double ground design" compared to the "single ground design" of the others. It is claimed by the authors that "the coupling between the DC layer and the DC lines from other unit-cells may deteriorate the metasurface performance." Please provide evidence (e.g., simulation results) for the above claim.

Response: We thank the reviewer for this comment. The evidence supporting the claim that "the coupling between the DC layer and the DC lines from other unit-cells may deteriorate the metasurface performance" is provided as follows.

Figure. R4 - 1 shows the DC-line network layout. We can see that DC lines potentially have different orientations and pass through multiple unit-cells. Thus, we studied by full-wave simulation the influences of the DC lines from other unit-cells on the performance of single-ground and double-ground designs, by considering two common scenarios encountered in the construction of the metasurface. One is the variation of the DC-line orientation, and the other is variation of DC-line numbers.

We first investigated the influence of the DC-line orientation on the unit-cell's performance. As can be seen from Figure. R4 - 2(a), (e), (i), and (m), the transmissive amplitudes of the single-ground unit-cell are affected by the DC-line orientation. As the angle θ increases from 0 to 30°, a dip appears in the corresponding amplitude curves ($\theta \neq 0$) around the operating frequency of 5.8 GHz. This can deteriorate the transmissive amplitudes and in turn deteriorate the system efficiency. In contrast, the DC-line orientation has little influence on the transmission of the double-ground unit-cell, as

shown in Figure. R4 - 2(c), (g), (k), and (o). The overlapping of all amplitude (and phase) curves suggests that the double-ground unit-cell is resistant to changes in the DC-line orientation. We also investigated the effect of the DC-line number on the unit-cell's performance, as depicted in Figure. R4 - 3. As shown in Figure. R4 - 3 (i), the transmissive amplitude decreases as the DC-line number increases. On the other hand, as shown in Figure. R4 - 3 (k), the double-ground unit-cell's transmissive amplitude remains stable regardless of the number of DC lines.

From the analysis presented above, it can be observed that the two-ground design exhibits robustness with respect to the orientation and number of DC lines. In the case of the single-ground design, the performance of the unit-cell can be compromised due to the coupling between the DC layer and DC lines. Conversely, the double-ground design effectively mitigates this coupling, due to the presence of two grounds. As a result, the double-ground unit-cell demonstrates robustness against variations in the DC-line orientation and number.

Figure. R4 - 1. The DC-line network layout.

Figure. R4 - 2. The simulated results of the unit-cells transmissive coefficients under different DC-line angles θ . The top panels are the simulation models of single- and double-ground unit-cells. The substrates are hidden for easy visibility.

Figure. R4 - 3. The simulated results of the unit-cells transmissive coefficients under different number of DC lines. The top panels are the simulation models of single- and double-ground unit-cells. The substrates are hidden for easy visibility.

REVIEWERS' COMMENTS

Reviewer #4 (Remarks to the Author):

Thank you for your effort. I have no further comment.

Reviewer #4 (Remarks to the Author):

Thank you for your effort. I have no further comment.

Responses to Reviewer #4:

Comment: Reviewer #4 (Remarks to the Author):

Thank you for your effort. I have no further comment.

Response: We thank the reviewer for the positive comment.